# Flow Snapshot Neurons in Action: Deep Neural Networks Generalize to Biological Motion Perception

**Shuangpeng Han**[1,2], **Ziyu Wang**[1,2,3], **Mengmi Zhang**[1,2]

[1]College of Computing and Data Science, Nanyang Technological University, Singapore
[2]Deep NeuroCognition Lab, Agency for Science, Technology and Research (A*STAR)
[3]Show Lab, National University of Singapore, Singapore
Address correspondence to mengmi.zhang@ntu.edu.sg

## Abstract

Biological motion perception (BMP) refers to humans' ability to perceive and recognize the actions of living beings solely from their motion patterns, sometimes as minimal as those depicted on point-light displays. While humans excel at these tasks *without any prior training*, current AI models struggle with poor generalization performance. To close this research gap, we propose the Motion Perceiver (MP). MP solely relies on patch-level optical flows from video clips as inputs. During training, it learns prototypical flow snapshots through a competitive binding mechanism and integrates invariant motion representations to predict action labels for the given video. During inference, we evaluate the generalization ability of all AI models and humans on 62,656 video stimuli spanning 24 BMP conditions using point-light displays in neuroscience. Remarkably, MP outperforms all existing AI models with a maximum improvement of 29% in top-1 action recognition accuracy on these conditions. Moreover, we benchmark all AI models in point-light displays of two standard video datasets in computer vision. MP also demonstrates superior performance in these cases. More interestingly, via psychophysics experiments, we found that MP recognizes biological movements in a way that aligns with human behaviors. Our data and code are available at link.

## 1 Introduction

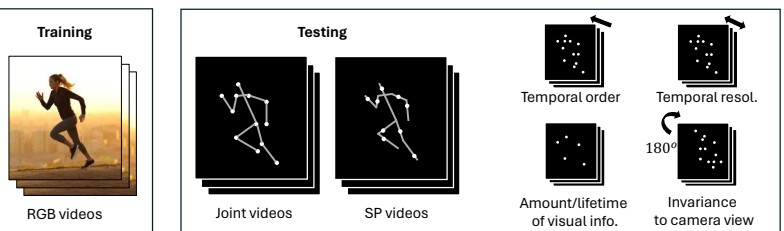

Figure 1: **Humans excel at biological motion perception (BMP) tasks with *zero* training, while current AI models struggle with poor generalization performance.** AI models are trained to recognize actions from natural RGB videos and tested using BMP stimuli on point-light displays, which come in two forms: Joint videos, which display only the detected joints of actors in white dots, and Sequential position actor videos (SP), where light points in white are randomly positioned between joints and reallocated to other random positions on the limb in subsequent frames (**Sec. 3.1**). Note that skeletons, shown in gray in the example video, are not visible to humans or AI models during testing. The generalization performance of both humans and models is assessed after varying five properties in temporal and visual dimensions. See **Appendix, Sec. A.1** for example videos.

38th Conference on Neural Information Processing Systems (NeurIPS 2024).

Biological Motion Perception (BMP) refers to the remarkable ability to recognize and understand the actions and intentions of other living beings based solely on their motion patterns [33]. BMP is crucial for tasks such as predator detection [24], prey selection [23], courtship behavior [69], and social communications [18, 2, 1] among primates and humans. In classical psychophysical and neurophysiological experiments [45], motion patterns can sometimes be depicted minimally, such as in point-light displays where only major joints of human or animal actors are illuminated. Yet, *without any prior training*, humans can robustly and accurately recognize actions [6, 71, 67] and characteristics of these actors like gender [53, 78], identity [16, 73], personalities [7], emotions [20], social interactions [93], and casual intention [77].

To make sense of these psychophysical and neurophysiological data, numerous studies [33, 34, 55, 89, 9, 52] have proposed computational frameworks and models in BMP tasks. Unlike humans, who excel at BMP tasks without prior training, these computational models are usually trained under specific BMP conditions and subsequently evaluated under different BMP conditions. However, the extent to which these models can learn robust motion representations from natural RGB videos and generalize them to recognize actions on BMP stimuli remains largely unexplored.

In parallel to the studies of BMP in psychology and neuroscience, action recognition on images and videos in computer vision has evolved significantly over the past few decades due to its wide range of real-world applications [15, 41, 81, 82, 57, 74]. The field has progressed from relying heavily on hand-crafted features [99, 75, 54] to employing deep-learning-based approaches [90, 95, 11, 101, 28, 30, 29, 25, 4, 94]. These modern approaches can capture the temporal dynamics and spatial configurations of complex activities within dynamic and unstructured environments [80, 92]. Despite these advancements, existing AI models still struggle with generalization issues related to occlusion [103, 3, 58], noisy environments [109], viewpoint variability [27, 61, 100], subtle human movements [83, 44] and appearance-free motion information [42]. While various solutions have been proposed to enhance AI generalization in action recognition [12, 72, 14, 70, 104, 48, 59, 113, 79, 50], they do not specifically tackle the generalization challenges in BMP tasks.

Here, our objective is to systematically and quantitatively examine the generalization ability of AI models trained on natural RGB videos and tested in BMP tasks. To date, research efforts in neuroscience and psychology [44, 34, 37] have mostly focused on specific stimuli for individual BMP tasks. There is a lack of systematic, integrative, and quantitative exploration that covers multiple BMP properties and provides a systematic benchmark for evaluating both human and AI models in these BMP tasks. To bridge this gap, we establish a benchmark BMP dataset, containing 62,656 video stimuli in 24 BMP conditions, covering 5 fundamental BMP properties. Our result indicates that current AI models exhibit limited generalization performance, slightly surpassing chance levels.

Subsequently, to enhance the generalization capability of AI models, we draw inspiration from [33] in neuroscience and introduce Motion Perceiver (MP). MP only takes dense optical flows between any pairs of video frames as inputs and predicts action labels for the given video. In contrast to many existing pixel-level optical flow models [90, 91, 86, 99], MP calculates dense optical flows at the granularity of patches from the feature maps. In MP, we introduce a set of flow snapshot neurons that learn to recognize and store prototypical motion patterns by competing with one another and binding with dense flows. This process ensures that similar movements activate corresponding snapshot neurons, promoting consistency in motion pattern recognition across patches of video frames. The temporal dynamics within dense flows can vary significantly depending on factors such as the speed, timing, and duration of the actions depicted in video clips. Thus, we also introduce motion invariant neurons. These neurons decompose motions along four motion directions and integrate their magnitudes over time. This process ensures that features extracted from dense flows remain invariant to small changes and distortions in temporal sequences.

We conducted a comparative analysis of the generalization performance of MP against existing AI models in BMP tasks. Impressively, MP surpasses these models by 29% and exhibits superior performance in point-light displays of standard video datasets in computer vision. Additionally, we examined the behaviors of MP alongside human behaviors across BMP conditions. Interestingly, the behaviors exhibited by MP in various BMP conditions closely align with those of humans. Our main contributions are highlighted:

**1.** We introduce a comprehensive large-scale BMP benchmark dataset, covering 24 BMP conditions and containing 62,656 video stimuli. As an upper bound, we collected human recognition accuracy in BMP tasks via a series of psychophysics experiments.

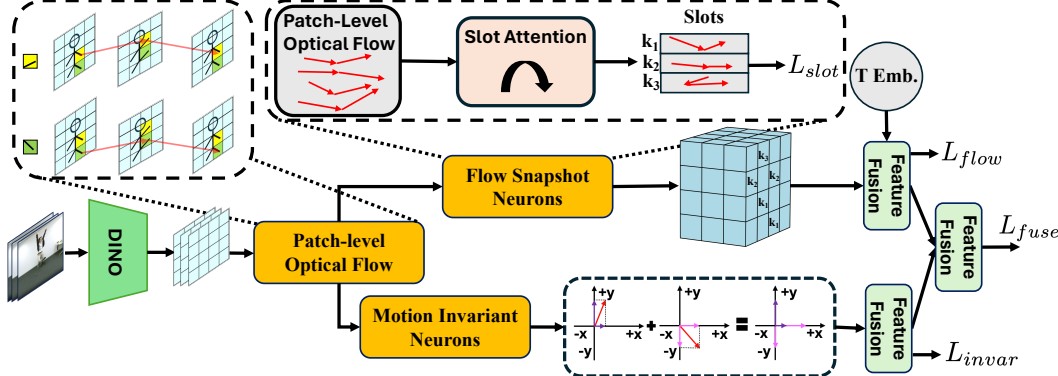

Figure 2: **Architecture of our proposed Motion Perceiver (MP) model.** Given a reference patch (yellow or green example patches), MP computes its patch-level optical flow (red arrows, **Sec. 2.1**) on the feature maps extracted from DINO [10]. Subsequently, these flows are processed through flow snapshot neurons **(Sec. 2.2)** and motion invariant neurons **(Sec. 2.3)** in two pathways. Activations from both groups of neurons are then integrated for action classification **(Sec. 2.4)**. Time embeddings (T Emb.) are used in the feature fusion process.

**2.** We propose the Motion Perceiver (MP). The model takes only patch-level optical flows of videos as inputs. Key components within MP, including flow snapshot neurons and motion invariant neurons, significantly enhance its generalization capability in BMP tasks.

**3.** Our MP model outperforms all existing AI models in BMP tasks, achieving up to a 29% increase in top-1 action recognition accuracy, and demonstrating superior performance in point-light displays of two standard video datasets in computer vision.

**4.** The behaviors exhibited by the MP model across various BMP tasks demonstrate a high degree of consistency with human behaviors in the same tasks. Network analysis within MP unveils crucial insights into the underlying mechanisms of BMP, offering valuable guidance for the development of generalizable motion perception capabilities in AI models.

## 2 Our Proposed Motion Perceiver (MP)

Our proposed model, Motion Perceiver (MP), aims to learn robust action-discriminative motion features from natural RGB videos and generalize these capabilities to recognize actions from BMP stimuli on point-light displays. MP takes the inputs of only patch-level optical flows from a video $\mathcal{V}$, which comprises $T$ frames denoted as $\{I_1, I_2, ..., I_t, ..., I_T\}$. While visual features from videos are typically useful for action recognition, our research focuses on extracting and learning motion information alone from natural RGB videos. Finally, MP outputs the predicted actions from a predefined set of labels $\mathcal{Y}$. See **Fig. 2** for the model architecture.

### 2.1 Patch-level Optical Flow

Pixel-level optical flow has been a common approach for modelling temporal dynamics in videos [90, 91, 86, 99]. Different from these works, we use the frozen ViT [21], pre-trained on ImageNet [19] with DINO [10], to extract feature map $F_t \in \mathbb{R}^{N \times C}$ from $I_t$ and compute its optical flow relative to other frames. Consider $F_t$ as a 2D grid of patches in $N = H \times W$, where $H$ and $W$ are the height and width of $F_t$. $N$ represents the number of patches in the feature map and $C$ is the feature dimension per patch. A unique spatial location for each patch can be defined by its 2D coordinates. We concatenate the X and Y coordinates of all patches in $F_t$ to form the patch locations $G \in \mathbb{R}^{N \times 2}$.

Two reasons motivate us to design patch-level optical flows. First, the empirical evidence [36] suggests that current unsupervised feature learning frameworks produce feature patches with semantically consistent correlations among neighboring patches. Therefore, patch-level optical flows convey meaningful motion information about semantic objects in a video. Second, pixel-level optical flows can be noisy due to motion blur, occlusions, specularities, and sensor noises. Feature maps obtained from feed-forward neural networks often mitigate these low-level perturbations and provide more accurate estimations of optical flows.

Next, we introduce how the patch-level optical flow for video $\mathcal{V}$ is computed. Without loss of generality, given any pair of feature sets $a \in \mathbb{R}^{m \times d}$ and $b \in \mathbb{R}^{n \times d}$, where $m$ and $n$ are the numbers of patches in the feature sets and $d$ is the feature dimension, the adjacency matrix $Q_{a,b}$ between $a$ and $b$ can be calculated as their normalized pairwise feature similarities: $Q_{(a,b)} = \frac{e^{f(a)f(b)^{\top}/\tau}}{\sum_n e^{f(a)f(b)^{\top}/\tau}} \in \mathbb{R}^{m \times n}$,

where the superscript $\top$ is the transpose function, $f(\cdot)$ is the $l_2$-normalization, and $\tau$ is the temperature controlling the sharpness of distribution with its smaller values indicating sharper distribution. We set temperature $\tau = 0.001$ and the influence of $\tau$ is analyzed in **Appendix, Tab. S4**.

Using the patch features of $I_t$ as the reference, the optical flow between any consecutive frames $I_m$ and $I_{m+1}$ is defined as $O^{I_t}_{m \to m+1} \in \mathbb{R}^{N \times 2}$.
$$O^{I_t}_{m \to m+1} = \hat{G}_{I_t \to I_{m+1}} - \hat{G}_{I_t \to I_m}, \text{ where } \hat{G}_{I_i \to I_j} = Q_{(F_i, F_j)}G. \tag{1}$$

$\hat{G}_{I_i \to I_j}$ represents the positions of all patches from $F_i$ of $I_i$ after transitioning to $F_j$ of $I_j$. Essentially, the positions of patches with shared semantic features on $F_i$ tend to aggregate towards the centroids of corresponding semantic regions on $F_j$. Consequently, the optical flow $O^{I_t}_{m \to m+1}$ indicates the movement from $I_m$ to $I_{m+1}$ for patches that exhibit similar semantic features as those in $I_t$.

For all $m \in \{1, 2, ..., T-1\}$ and $t \in \{1, 2, ..., T\}$, we compute $O^{I_t}_{m \to m+1}$ and concatenate them to obtain the patch-level optical flow $\hat{O} \in \mathbb{R}^{T \times N \times 2 \times (T-1)}$ for video $\mathcal{V}$ as $\hat{O} = [O^{I_1}_{1 \to 2}, ..., O^{I_T}_{(T-1) \to T}]$.

Since small optical flows might be susceptible to noise or errors, the optical flows in $\hat{O}$ with magnitudes less than $\gamma = 0.2$ are set to $10^{-6}$ to maintain numerical stability. Unlike [90] where optical flows are computed only between two adjacent frames, our method introduces $\hat{O}$, where we compute a sequence of optical flows across all $T$ frames for any reference patch of a video frame. This dense optical flow estimation approach at the patch level captures a richer set of motion dynamics, providing a more comprehensive analysis of movements throughout the video.

## 2.2 Flow Snapshot Neurons

Drawing on findings from neuroscience that highlight biological neurons selective for complex optic flow patterns associated with specific movement sequences in motion pathways [33], we introduce "flow snapshot neurons". These neurons are designed to capture and represent prototypical moments or "slots" in movement sequences. Mathematically, we define $K = 6$ flow snapshot neurons or slots $\hat{Z} \in \mathbb{R}^{K \times D}$, where $D = 2 \times (T-1)$ is the feature dimension per slot. $\hat{Z}$ contains learnable parameters randomly initialized with the Xavier uniform distribution [35]. The impact of $K$ is discussed in **Appendix, Tab. S4**.

Slot attention mechanism [63] separates and organizes different elements of an input into a fixed number of learned prototypical representations or "slots". The learned slots have been useful for semantic segmentation [102, 60, 111, 51, 22]. Here, we follow the training paradigm and implementations of the slot attention mechanism in [63] and apply it to $\hat{O}$. Specifically, each slot in $\hat{Z}$ attends to unique optical flow sequence patterns in $\hat{O}$ through a competitive attention mechanism based on feature similarities. To ensure the prototypical optical flow patterns in $\hat{Z}$ are diverse, we introduce the loss of contrastive walks $L_{slot}$ [102]. During inference, we keep $\hat{Z}$ frozen and leverage the activations of flow snapshot neurons for action recognition, denoted as $\hat{M} = f(\hat{O})f(\hat{Z})^{\top}$, where $\hat{M} \in \mathbb{R}^{T \times N \times K}$. See **Appendix, Sec. B.1** and **Appendix, Sec. B.2** for mathematical formulations.

Unlike [108] that use slot-bonded optical flow sequences $\hat{M}\hat{Z} \in \mathbb{R}^{T \times N \times D}$ for downstream tasks, we highlight two advantages of using $\hat{M}$. First, the dimensionality of flow similarities ($K$) is typically much smaller than that of slot-bonded optical flow sequences ($D$), effectively addressing overfitting concerns and reducing computational overhead. Second, by leveraging flow similarities, we benefit from slot activations that encode prototypical flow patterns distilled from raw optical flow sequences. This filtration process helps eliminate noise and irrelevant information from the slot-bonded flow sequences, enhancing the model's robustness.

## 2.3 Motion Invariant Neurons

The temporal dynamics in video clips can vary significantly in the speed and temporal order of the actions. Here, we introduce motion invariant neurons that capture the accumulative motion magnitudes independent of frame orders. Specifically, we define the optical flow $(O^{x,n,I_t}_{m \to t}, O^{y,n,I_t}_{m \to t})$

for patch $n$ in $I_m$ moving from frame $I_m$ to $I_t$ along $x$ and $y$ axes. Every patch-level optical flow in $\hat{O}$ can be projected into four motion components along $+x$, $-x$, $+y$ and $-y$ axes. Without loss of generality, we show that for patch $n$ in $I_t$, all its optical flow motions over $T$ frames along the $+x$ axis are aggregated:

$$\frac{1}{T} \sum_{m=1}^{T} O_{m \to t}^{x,n,I_t} \mathbf{1}_{O_{m \to t}^{x,n,I_t} > 0} \text{ where } \mathbf{1}_{u>0} = \begin{cases} 1 & \text{if } u > 0, \\ 0 & \text{otherwise.} \end{cases} \quad (2)$$

By repeating the same operations along $-x, +y, -y$ axes for all the $N$ patches over $T$ frames, we obtain the motion invariant matrix $\tilde{M} \in \mathbb{R}^{T \times N \times 4}$. A stack of self-attention blocks followed by a global average pooling layer fuse information along 4 motion components and then along temporal dimension $T$. To encourage the model to learn motion invariant features for action recognition, we introduce cross-entropy loss $L_{invar}$ to supervise the predicted action against the ground truth $\mathcal{Y}$.

## 2.4 Multi-scale Feature Fusion and Training

Processing videos at multiple temporal resolutions allows the model to capture both rapid motions and subtle details, as well as long-term dependencies and broader contexts. We use the subscript in $\hat{O}$, $\hat{Z}$, and $\hat{M}$ to indicate the temporal resolution. Instead of computing $\hat{O}_1$ between consecutive frames only, we increase the stride sizes to 2, 4, and 8. For example, $\hat{O}_4 \in \mathbb{R}^{T \times N \times 2 \times (T/4-1)}$ denotes patch-level flows computed between every 4 frames. Note that we maintain a stride size of 1 for $\tilde{M}$ to ensure motion invariance to temporal orders across the entire video, as subsampling frames would not adequately capture this attribute.

For action recognition, the activations of flow snapshot neurons are concatenated across multiple temporal resolutions as $\hat{M}_{1,2,4,8} = [\hat{M}_1, \hat{M}_2, \hat{M}_4, \hat{M}_8]$ and then used for the fusion of motion information. The concatenated data is first processed through a series of self-attention blocks that operate across $4K$ slot dimensions, followed by a global average pooling over these dimensions. We then repeat the same fusion process over $T$ time steps. Time embeddings, similar to the sinusoidal positional embeddings in [98], are applied across frames. These embeddings help incorporate temporal context into the learned features. The resulting integrated motion feature vector is used for action recognition, with a cross-entropy loss $L_{flow}$ to supervise the predicted action against $\mathcal{Y}$.

While $\hat{M}_{1,2,4,8}$ captures the detailed temporal dynamics, $\tilde{M}$ learns robust features against variations in temporal orders. MP combines feature vectors from $\hat{M}_{1,2,4,8}$ and $\tilde{M}$ with a fully connected layer and outputs the final action label. A cross-entropy loss $L_{fuse}$ is used to balance the contributions from $\hat{M}_{1,2,4,8}$ and $\tilde{M}$. The overall loss is: $L = \alpha L_{slot} + L_{flow} + L_{invar} + L_{fuse}$, where the loss weight $\alpha = 10$. See **Appendix, Tab. S4** for the effect of $\alpha$.

**Implementation Details.** Our model is trained on Nvidia RTX A5000 and A6000 GPUs, and optimized by AdamW optimizer [65] with cosine annealing scheduler [64] starting from the initial learning rate $10^{-4}$. Data loading is expedited by FFCV [56]. All videos are downsampled to $T = 32$ frames. We use a random crop of $224 \times 224$ pixels with horizontal flip for training, and a central crop of $224 \times 224$ pixels for inference. We use the same set of hyper-parameters for our model in all the datasets. More training details are in **Appendix, Sec. C**.

## 3 Experiments

### 3.1 Our Biological Motion Perception (BMP) Dataset with Human Behavioral Data

Following the works in vision science, psychology, and neuroscience [33, 89, 9, 53, 6, 16, 37, 45, 71, 76, 97], we introduce the BMP dataset, comprising 10 action classes (**Appendix, Sec. A.2**) specifically chosen for their strong temporal dynamics and minimal reliance on visual cues. Studies [107, 112, 32, 85] indicate that current action recognition models often rely on static features. Point-light displays reduce confounding factors, such as colors, sketches, and body limbs, by minimizing visual information, thereby highlighting the ability to perceive motion. This selection criterion ensures that specific objects or scene contexts, such as a soccer ball on green grass, do not bias the AI models toward recognizing the action as "playing soccer". This approach aligns with our research focus on generalization in motion perception. In the BMP dataset, there are three types of visual stimuli.

**Natural RGB videos (RGB):** We incorporated 9,492 natural RGB videos from the NTU RGB+D 120 dataset [62] and applied a 7-to-3 ratio for dividing the data into training and testing splits. **Joint**

**videos (J):** We applied Alphapose [26] to identify human body joints in the test set of our RGB videos. The joints of a moving human are displayed as light points, providing minimal visual information other than the joint positions and movements of these joints over time. **Sequential position actor videos (SP):** SP videos are generated to investigate scenarios where local inter-frame motion signals are eliminated [6]. Light points are positioned randomly between joints rather than on the joints themselves. In each frame, every point is relocated to another randomly selected position on the limb between the two joints with uniform distribution. This process ensures that no individual point carries the valid local image motion signal of limb movement. Nonetheless, the sequence of static postures in each frame still conveys information about body form and motion.

We investigated 5 fundamental properties of motion perception by manipulating various design parameters of these stimuli. We adopted the experiment naming convention $[type] + [condition]$ to denote the experimental setup for the stimulus types and the specific manipulation conditions applied to them. For instance, [Joint-3-Frames] indicates that three frames are uniformly sampled from the Joint video type. Next, we introduce the fundamental properties of motion perception and the manipulations performed to examine these properties.

**Temporal order (TO):** To disrupt the temporal order [84], we reverse (Reversal) or randomly shuffle (Shuffle) video frames for two types of videos, RGB and Joint above. **Temporal resolution (TR):** We alter the temporal resolution [84] by uniformly downsampling 32 frames to 4 or 3 frames, labelled as 4-Frames and 3-Frames, respectively. For models that need a fixed frame count, each downsampled frame is replicated multiple times before advancing to the next frame in the sequence, until we reach the necessary quantity. **Amount of visual information (AVI):** We quantify the amount of visual information based on the number of light points (P) in Joint videos [45]. Specifically, we included conditions: 5, 6, 10, 14, 18, and 26 light points. **Lifetime of visual information (LVI):** In SP videos, we manipulate the number of consecutive frames during which each point remains at a specific limb position before being randomly reassigned to a different position. Following [6], we refer to this parameter as the "lifetime of visual information" (LT), and include LT values of 1, 2, and 4. A longer lifetime implies that each dot remains at the same limb position for a longer duration, thereby conveying more local image motion information. **Invariance to camera views (ICV):** Neuroscience research [43] has revealed brain activity linked to decoding both within-view and cross-view action recognition. To probe view-dependent generalization effects, we sorted video clips from Joints into three categories based on camera views: frontal, $45°$, and $90°$ views. See **Appendix, Sec. A.3** for more implementation details.

**Human psychophysics experiments:** We conducted human psychophysics experiments schematically illustrated in **Appendix, Sec. D**, using Amazon Mechanical Turk (MTurk) [8]. A total of 90 subjects were recruited. For data quality controls, we implemented checks during the experiment. Following a set of filtering criteria, we retained data from 88 subjects. See **Appendix, Sec. D** for details.

### 3.2 Video Action Recognition Datasets and Baselines in Computer Vision

To evaluate the ability of all AI models to recognize a wider range of actions in natural RGB videos, we include two standard video action recognition datasets in computer vision. **NTU RGB+D 60** [87] comprises 56,880 video clips featuring 60 action classes. **NW-UCLA [100]** contains 1,494 videos featuring 10 action classes. Both datasets are split randomly, with a 7:3 ratio for the training and test sets. All AI models are trained on RGB videos and tested on Joint videos of these datasets curated with Alphapose [26] described above. We compute the top-1 action recognition accuracy of all AI models. In addition to our MP model, we include six competitive baselines below for comparison: **ResNet3D [38]** adapts the architecture of ResNet [39] by substituting the 2D convolutions with the 3D convolutions. **I3D [11]** extends a pre-trained 2D-CNN on static images to 3D by duplicating 2D filters along temporal dimensions and fine-tuning on videos. **X3D [28]** expands 2D-CNN progressively along spatial and temporal dimensions with the architecture search. **R(2+1)D [96]** factories the 3D convolution kernels into spatial and temporal kernels. **SlowFast [29]** is a two-stream architecture that processes temporal and spatial cues separately. **MViT [25]** is a transformer-based model that expands feature map sizes along the temporal dimension while reducing them along the spatial dimension. Furthermore, we include three more competitive baselines (E2-S-X3D [42], VideoMAE [94], TwoStream-CNN [90]) and present results in **Appendix, Sec. E**. The findings remain consistent with the baselines above. All baselines except for TwoStream-CNN are pre-trained on Kinetics 400 [47]. TwoStream-CNN is pre-trained on ImageNet [19]. As an upper bound, we also train the MP directly on Joint or SP videos and the results are in **Appendix, Sec. E**.

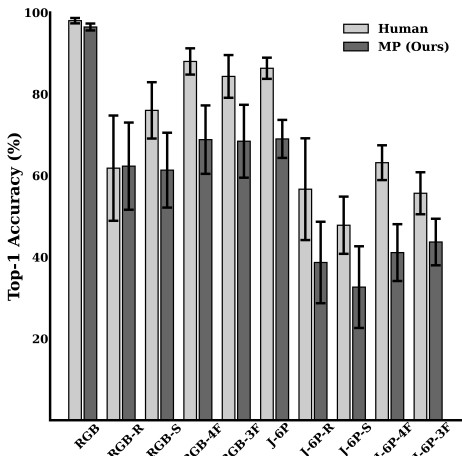 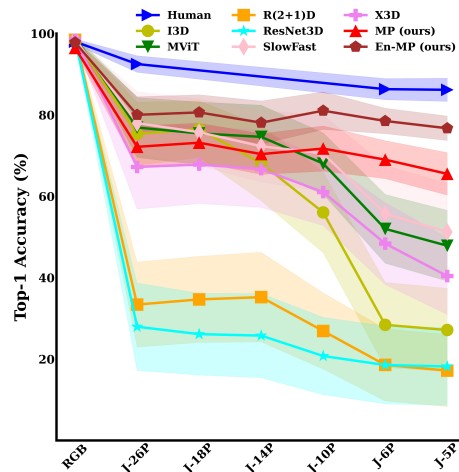

Figure 3: **Temporal orders and resolutions matter in generalization performance on RGB and Joint videos.** Stimuli encompass RGB and Joint (J) videos. Short forms include R (reversal), S (shuffle), F (frames), and P (points) in **Sec. 3.1**. Error bars indicate the standard error of the top-1 accuracy across different action classes.

Figure 4: **Our model demonstrates human-like robustness under reduced visual information.** Top-1 action recognition accuracy is a function of the number of points (P) in Joint (J) videos. Results from RGB test videos are at the leftmost. The colored shaded region represents the standard error across all action classes.

## 4    Results

### 4.1    Our model achieves human-level performance without task-specific retraining

We explore five key properties of motion perception and their influence on video action recognition tasks between humans and our model (MP). Detailed comparisons between humans and all AI models across all BMP tasks are provided in the **Appendix, Sec. L**.

**Temporal order and temporal resolution matter more in Joint videos than natural RGB videos.** In **Fig. 3**, both humans and MP exhibit higher top-1 accuracy in RGB videos (RGB) compared to Joint videos (J-6P) due to the increased difficulty of generalization in Joint videos from their minimal motion and visual information. Error bars in **Fig. 3** represent the standard error of top-1 accuracy across different action classes. However, despite this challenge, the performance on Joint videos remains significantly above chance (1/10). This suggests that both humans and MP have remarkable abilities to recognize actions based solely on their motion patterns in point-light displays.

Moreover, we observed that shuffled (S) or reversed (R) temporal orders significantly impair recognition performance in both humans and MP. This effect is more pronounced in Joint videos (J-6P-R and J-6P-S) compared to RGB videos (RGB-R and RGB-S). The minimal motion information available in Joint videos makes them particularly susceptible to disruptions in temporal orders. The same behavioral patterns are also captured by MP. Interestingly, shuffling has a lesser impact on human performance compared to reversing RGB videos (RGB-R versus RGB-S, p-value < 0.05). However, this pattern is reversed in Joint videos (J-6P-R versus J-6P-S, p-value < 0.05). When considering actions such as sitting down versus standing up, reversing orders may alter the temporal context in RGB videos. The effect of temporal continuity outweighs the effect of temporal context in Joint videos.

We conjectured that temporal resolution matters in video action recognition. Indeed, a decrease in top-1 accuracy with decreasing temporal resolutions is observed in both RGB and Joint videos (compare 32, 4, and 3 frames). However, this effect is more pronounced in Joint videos (J-6P, J-6P-4F, J-6P-3F) compared to RGB videos (RGB, RGB-4F, RGB-3F). Surprisingly, even in the most challenging J-6P-3F, both humans and MP achieve top-1 accuracy of over 40%, significantly above chance. This suggests that both humans and MP are robust to changes in temporal resolutions.

**Minimal amount of visual information is sufficient for recognition.** In **Fig. 4**, both humans and all AI models exhibit comparable performances in RGB videos (RGB). Interestingly, the accuracy drop from J-6P to J-5P is indistinguishable for humans (p-value > 0.05). Similarly, there was a wide range of the number of points that led to robust action recognition for MP (J-26P to J-5P). However,

its absolute accuracy is much lower than humans. Therefore, we introduce an enhanced version of the MP model (En-MP) by extending it across multiple DINO blocks, in addition to the last DINO block. Detailed implementation of the En-MP is provided in **Appendix, Sec. F**. Surprisingly, the En-MP outperforms our original MP significantly and performs competitively well as humans. These observations suggest that a minimal visual representation in J-5P is sufficient for action recognition on humans and our En-MP. Conversely, traditional AI models show a significant performance decline moving from J-26P to J-5P. These findings suggest that existing AI models struggle with reduced visual information.

**Humans and MP are robust to the disrupted local image motion.** Aligned with [6, 76], in **Fig. 5** top-1 accuracy in SP videos improves with an increased number of points for humans, a trend that MP replicates. Interestingly, both humans and MP do not show obvious increased performance with lifetimes of points more than 1 frame, potentially due to the loss of form information from fewer dots. This indicates that humans and MP capture not just local motions but also dynamic posture changes.

**Camera views have minimal impacts on recognition.** Neuroscience studies [43, 97, 46, 68] provide evidence that both viewpoint-dependent and viewpoint-invariant representations for action recognition are encoded in primate brains. In **Fig. 7**, MP exhibits a slightly higher accuracy on frontal and $45°$ views compared to the $90°$ (profile) view by 2.5%, which mirrors human performance patterns in [46]. However, we note that human behaviors in our study differ slightly from [46] (see **Appendix, Sec. G**). Moreover, MP significantly outperforms the second-best model, MViT, by 18%. This suggests that MP not only captures human-like viewpoint-dependent representations but also maintains superior accuracy among AI models across different camera views.

## 4.2 Comparisons among AI models in BMP tasks and standard computer vision datasets

**MP aligns with human behaviors more closely than all the competitive baselines in BMP tasks.** In **Fig. 6**, we reported the correlation coefficients between each AI model and human performance across all conditions in the BMP dataset. The results demonstrate that our MP exhibit a significantly higher correlation with human performance compared to the baselines, indicating that our MP align more closely with human performance. In addition to the correlation coefficients, we also present the error pattern consistency [31] between models and humans (**Appendix, Sec. H**, **Appendix, Fig. S4**). Results suggest that our En-MP achieves the highest error pattern consistency with humans at the trial level.

**MP significantly outperforms all the competitive baselines in BMP tasks.** In **Appendix, Fig. S3**, we present the absolute accuracy of all models and humans. The slopes near 1 in our MP model indicate that it performs on par with humans and surpasses all baselines across the five BMP dataset properties. Despite explicitly modelling motion and visual information in separate streams, the SlowFast model struggles with temporal order. Likewise, MViT, which leverages transformer architectures, fails to generalize across different temporal resolutions.

**MP outperforms all the competitive baselines on Joint videos from two standard computer vision datasets.** In **Tab. 1**, MP, relying solely on patch-level optical flows as inputs, performs above chance. This suggests that motion information alone is sufficient for accurate action recognition. Moreover, MP performs better than existing models on Joint videos in these datasets, highlighting that MP learns to capture better motion representations from RGB videos during training. In **Appendix, Sec. I**, we provide results and discussions when we explicitly feed pixel-level optical flows as inputs to these baselines during training and test these models on the Joint videos. Our results show that MP still outperforms these models, suggesting that MP learns generalizable motion representations beyond optical flow patterns. In **Appendix, Sec. J**, we also present visualizations of patch-level optical flow examples, demonstrating its ability to semantically segment the movements of a person.

## 4.3 Ablation studies reveal key components in our model

To investigate how different components of MP contribute to the generalization performance on Joint and SP videos, we conducted ablation studies (**Tab. 2**, **Appendix, Tab. S4** and **Appendix, Tab. S5**). Removing the pathway involving motion invariant neurons leads to a drop in accuracy when video frames are shuffled or reversed (**A1**, **Tab. 2** and **Appendix, Fig. S3**). For example, MP outperforms its ablated model without motion invariant neurons in the following experiments: 62.32% vs 49.58% in RGB-R; 61.34% vs 38.03% in RGB-S, 38.69% vs 36.03% in J-6P-R, and 32.65% vs 25.28% in J-6P-S. Similarly, removing the pathway involving flow snapshot neurons also leads to a significant

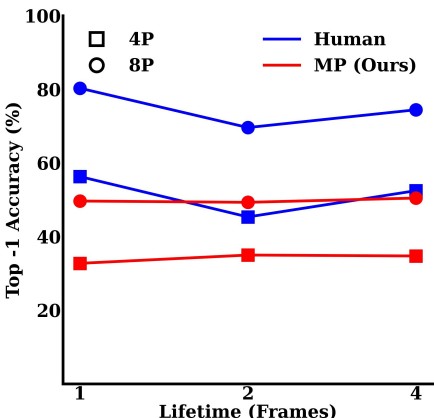

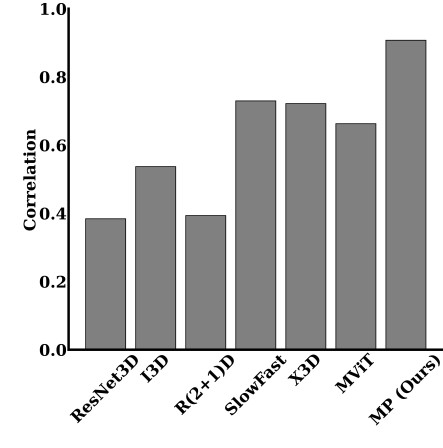

Figure 5: **Both humans and our model can recognize actions in SP videos without local motions.** Performance varies depending on the persistence of visual information, with stimuli having 4 and 8 points (P) of the actors (**Sec. 3.1**).

Figure 6: **Our MP model shows a significantly stronger correlation with human performance compared to all baselines.** The correlation between the model and human performance across all BMP conditions is presented.

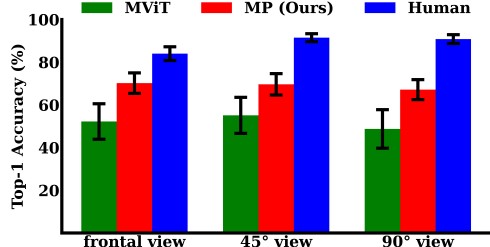

Figure 7: **Humans and AI models show minimal difference in generalization across camera views on J-6P videos (Sec. 3.1).** Error bars indicate the standard error.

Table 1: **Our model outperforms all existing models on Joint videos (J-6P) from two standard computer vision datasets.** See **Sec. 3.2** for dataset descriptions. Best is in bold.

| Method | Top-1 Acc. (%) | |
| --- | --- | --- |
| | NTU RGB+D 60 | NW-UCLA |
| ResNet3D | 6.89 | 10.99 |
| I3D | 4.46 | 8.74 |
| X3D | 2.00 | 16.59 |
| R(2+1)D | 5.18 | 10.54 |
| SlowFast | 6.26 | 10.09 |
| MViT | 6.56 | 10.09 |
| MP (ours) | **20.38** | **42.83** |

accuracy drop (**A2**). Both ablation studies highlight the importance of two pathways for action recognition.

Instead of multi-scale feature fusion, we utilize a single-scale pathway of flow snapshot neurons (**A3**). The decrease of 10.57% in accuracy on J-6P videos implies that multi-scale flow snapshot neurons capture both fine-grained and high-level temporal patterns, essential for action recognition. In **A4**, the model takes patch-level optical flows as inputs and directly utilizes them for feature fusion and action classification without flow snapshot neurons. There is a significant drop of 14.33% in accuracy, implying that flow snapshot neurons capture meaningful prototypical flow patterns, and their activations are critical for recognition. Threshold $\gamma$ controls the level of noise or errors in small motions. As expected, removing $\gamma$ leads to impaired recognition accuracy (**A5**).

Data augmentation is a common technique to enhance AI model generalization [88, 110, 17, 40, 114, 49]. It typically includes temporal augmentations like random shuffling and reversal in **A6**. Surprisingly, MP's generalization performance is impaired, indicating that randomizing frame orders during training disrupts motion perception and action understanding. We also present ablation results in RGB videos, with similar conclusions drawn, albeit to a lesser extent due to increased visual information and reduced reliance on motion.

In addition to the main ablation results discussed here, we also provide the extra ablation studies and model analysis in the **Appendix**. Specifically, we include the following studies: 1. the removal of the time embedding in our MP (**Sec. 2.4, Appendix, Tab. S5**); 2. the removal of the loss term $L_{slot}$ in our MP (**Sec. 2.2, Appendix, Tab. S5**); 3. the choice of the reference frame for calculating path-level optical flows in our MP (**Sec. 2.1, Appendix, Tab. S5**); 4. the pixel-level optical flows downscaled to the size of the patch-level optical flows in our MP (**Appendix, Sec. K**); 5. the replacement of

| Ablation ID | P-MIN | P-FSN | SS FSN | Slots $\hat{Z}$ | Thres. $\gamma$ | T Aug. | Top-1 Acc. (%) | | |
|---|---|---|---|---|---|---|---|---|---|
| | | | | | | | RGB | J-6P | SP-8P-1LT |
| 1 | ✗ | ✓ | ✗ | ✓ | ✓ | ✗ | 93.47 | 64.54 | 49.23 |
| 2 | ✓ | ✗ | ✗ | ✗ | ✗ | ✗ | 86.17 | 42.52 | 25.91 |
| 3 | ✓ | ✗ | ✓ | ✓ | ✓ | ✗ | 96.07 | 58.43 | 30.83 |
| 4 | ✓ | ✓ | ✗ | ✗ | ✓ | ✗ | **96.84** | 54.67 | 35.43 |
| 5 | ✓ | ✓ | ✗ | ✓ | ✗ | ✗ | 96.70 | 42.38 | 29.42 |
| 6 | ✓ | ✓ | ✗ | ✓ | ✓ | ✓ | 93.33 | 58.32 | 31.95 |
| Full Model (ours) | ✓ | ✓ | ✗ | ✓ | ✓ | ✗ | 96.45 | **69.00** | **49.68** |

Table 2: **Ablation reveals critical components in our model.** Top-1 accuracy is reported on RGB videos, Joint videos with 6 points (J-6P), and SP videos with 8 points and a lifetime of 1 (SP-8P-1LT). From left to right, the ablated components are: the pathway with Motion Invariant Neurons (P-MIN), the pathway with Flow Snapshot Neurons (P-FSN), the single-scale branch with Flow Snapshot Neurons $\hat{M}_1$ (SS FSN), Slots $\hat{Z}$, threshold $\gamma$ (**Sec. 2.1**), and data augmentation by randomly shuffling and reversing training frames within the same video. Best is in bold.

the feature extractor DINO with ResNet in our MP (**Appendix, Sec. K**); 6. the comparison of the number of trainable parameters among all the models (**Appendix, Sec. K**); 7. the analysis of key frames predicted by our MP on an example video in **Appendix, Sec. K**. All these studies emphasized the importance of our model designs and demonstrated the generalization ability of our model.

# 5 Discussion

We introduce Motion Perceiver (MP) as a generalization model trained on natural RGB videos, capable of perceiving and identifying actions of living beings solely from their minimal motion patterns on point-light displays, even without prior training on such stimuli. Within MP, flow snapshot neurons learn prototypical flow patterns through competitive binding, while motion invariant neurons ensure robustness to variations in temporal orders. The fused activations from both neural populations enable action recognition.

To evaluate the generalization capabilities of all AI models, we curated a dataset comprising 63k stimuli across 24 BMP conditions using point-light displays inspired by neuroscience. Psychophysics experiments on this dataset were conducted, providing human behavioral data for comparison with computational models. While AI models can surpass human performance in numerous tasks, current AI models for action recognition still fall short of human capabilities in many BMP conditions. By focusing solely on motion information, MP achieves superior generalization performance among all AI models and demonstrates a strong alignment with human behavioral responses. All baselines are pre-trained on large-scale video datasets, whereas our MP uses feature extractors pre-trained on naturalistic images. Remarkably, despite lacking video pre-training, our MP outperforms all baselines.

Our work takes an initial step toward bridging artificial and biological intelligence in BMP. First, it raises intriguing questions in neuroscience, such as the neural basis of motion invariant neurons, which are crucial when video frames are shuffled or reversed. Bio-inspired architectures can help test specific neuroscience hypotheses, while insights from neuroscience can, in turn, guide the design of more advanced AI systems. Second, our work paves the way for real-world applications requiring robust motion recognition, such as in low-light conditions where visual information is limited.

The 10 action classes in our BMP dataset were selected for their rich temporal information. We observe significant variations in action recognition accuracy across various action classes for both human participants and AI models. In the future, the BMP dataset could be expanded to include more complex actions. Both neuroscience and computer vision use various biological motion stimuli; here, we focus on point-light displays, but future work could explore other visual stimuli, such as motion patterns in noisy backgrounds. While our model shows promising generalization on BMP stimuli, it does not account for attention or top-down influences. Moreover, since it only uses patch-level optical flows, integrating information from the ventral (form perception) and dorsal (motion perception) pathways remains an open challenge. Further discussion on the social impact of our work can be found in **Appendix, Sec. M**.

## Acknowledgements

This research is supported by the National Research Foundation, Singapore under its AI Singapore Programme (AISG Award No: AISG2-RP-2021-025), and its NRFF award NRF-NRFF15-2023-0001. We also acknowledge Mengmi Zhang's Startup Grant from Agency for Science, Technology, and Research (A*STAR), Startup Grant from Nanyang Technological University, and Early Career Investigatorship from Center for Frontier AI Research (CFAR), A*STAR.

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

# A Biological Motion Perception (BMP) Dataset

## A.1 Example Videos in the BMP Dataset

We provide one example RGB video and its corresponding BMP videos under different conditions where the subject is performing a "sit down" action. The naming convention of these BMP videos follows the same as in **Sec. 3.1**. See the example video at the link.

## A.2 Action Classes in the BMP Dataset

There are 10 action classes in the BMP dataset: pick up, throw, sit down, stand up, kick something, jump up, point to something, nod head/bow, falling down and arm circles.

## A.3 More Implementation Details of the BMP Dataset

In **Sec. 3.1**, we vary the amount of visual information based on the number of point lights on Joint videos. To generate Joint videos, Alphapose [26] is used as the tool to detect the joints of a human body from RGB videos. After that, we use these detected joints to generate BMP stimulus on point-light displays. The specific positions of light points correspond to certain joint locations:

- **J-26P:** 1 point each on the head, nose, jaw, and abdomen, 1 point each on one eye, one ear, one shoulder, one elbow, one hand, one hip and one knee, and 8 points on two feet with 4 points on each foot;
- **J-18P:** 1 point each on the head, nose, jaw, and abdomen, 1 point each on one ear, one shoulder, one elbow, one hand, one hip, one knee and one ankle;
- **J-14P:** 1 point each on the nose, and abdomen, 1 point each on one shoulder, one elbow, one hand, one hip, one knee and one ankle;
- **J-10P:** 1 point each on the nose, and abdomen, 1 point each on one shoulder, one hand, one hip and one ankle;
- **J-6P:** 1 point each on the nose, and abdomen, 1 point each on one hand and one ankle;
- **J-5P:** 1 point each on the nose, 1 point each on one hand and one ankle.

We also looked into the invariance property to camera views in **Sec. 3.1**. The video clips in **J-6P** are categorized based on the viewpoints in the videos: the frontal view, $45°$ view and $90°$ view, which are respectively labelled as **J-6P-0V**, **J-6P-45V** and **J-6P-90V** in short form. $45°$ view and $90°$ view are rotated either clockwise or counterclockwise from the frontal view.

# B Mathematical Formulations of Flow Snapshot Neurons

## B.1 Slot Attention Module

As explained in **Sec. 2.2**, the slot attention module [63] aims to obtain flow snapshot neurons $\hat{Z} \in \mathbb{R}^{K \times D}$ to capture prototypical moments from the patch-level optical flow $\hat{O} \in \mathbb{R}^{S \times D}$ where $S = T \times N$ and $D = 2 \times (T - 1)$. Formally, based on the cross-attention mechanism [98], slots $\hat{Z}$ serve as the query, $\hat{O}$ contribute to both the key and the value, and $q(\cdot), k(\cdot), v(\cdot)$ are the linear transformation employed to project inputs and slots into a common dimension $B$, which can be formulated as:

$$\text{attn}_{i,j} := \frac{e^{J_{i,j}}}{\sum_{l=1}^{K} e^{J_{i,l}}} \quad \text{where} \quad J := \frac{1}{\sqrt{D}} k(\hat{O}) \cdot q(\hat{Z})^{\top} \in \mathbb{R}^{S \times K}, \tag{3}$$

$$h := U^{\top} \cdot v(\hat{O}) \in \mathbb{R}^{K \times B} \quad \text{where} \quad U_{i,j} := \frac{\text{attn}_{i,j}}{\sum_{l=1}^{S} \text{attn}_{l,j}}. \tag{4}$$

It is worth noting that the superscript $\top$ is the transpose function, and the attention matrix $U$ is normalized to make sure that no parts of the input are overlooked. Next, following the work [63] on slot attention, the slots are iteratively refined recurrently based on the Gated Recurrent Unit (GRU) [13] to maintain a smooth update. Let $\hat{Z}_0$ represent the initial slot base, with parameters

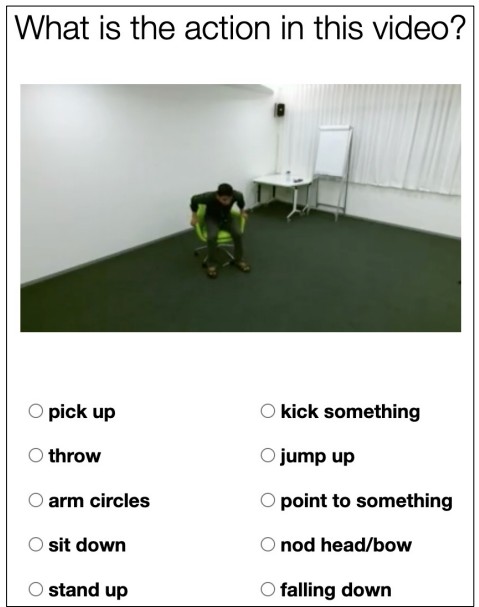

Figure S1: **Schematic of human psychophysics experiments on Amazon Mechanical Turk (MTurk).** In every trial, subjects are presented with a video and a list of ten options. After the video plays only once, the subject has to choose one action among ten options that best describe the action in the video.

initialized by randomly sampling from the Xavier uniform distribution [35]. We denote the hidden state at time step $p$ as $h_p$, and slots at time step $p$ and $p+1$ as $\hat{Z}_p$ and $\hat{Z}_{p+1}$. Next, we get:

$$\hat{Z}_{p+1} = \text{GRU}\left(\hat{Z}_p, h_p\right). \tag{5}$$

After $P$ iterations, the final slot features are represented as $\hat{Z} = \hat{Z}_P$. We empirically set $P = 3$ [63].

### B.2 Contrastive Walk Loss

To encourage diversity in prototypical optical flow patterns in $\hat{Z}$, we use the contrastive loss between $\hat{O}$ and $\hat{Z}$ introduced in [102].

$$L_{slot} = \text{CE}(Q_{(\hat{Z},\hat{O})}Q_{(\hat{O},\hat{Z})}, \mathbf{I}), \tag{6}$$

where $Q_{(a,b)}$ is defined as the normalized pairwise feature similarities between $a$ and $b$ in **Sec. 2.1**. We use the temperature $\mu$ of 0.05 in $Q_{(\hat{Z},\hat{O})}$ and $Q_{(\hat{O},\hat{Z})}$. CE$(\cdot,\cdot)$ stands for the cross-entropy loss and $\mathbf{I} \in \mathbb{R}^{K \times K}$ is the identity matrix. The effect of $\mu$ is covered in **Appendix, Tab. S4**.

## C   More Training Details

Our model is trained on the training sets of RGB videos in BMP, NTU RGBD+60 and NW-UCLA datasets respectively. **BMP:** Videos are resized to $224 \times 398$ pixels for 50 epochs training with a batch size of 48. **NTU RGB+D 60:** Videos are resized to $224 \times 398$ pixels and trained for 100 epochs with a batch size of 128. **NW-UCLA:** Videos are resized to $224 \times 300$ pixels and trained for 100 epochs using a batch size of 16.

## D   Human Psychophysics Experiments

**Fig. S1** demonstrates the schematic of human psychophysics experiments on Amazon Mechanical Turk (MTurk) [8]. We recruited a total of 90 human subjects. We collect data from a total of 12,600 trials.

|  | Top-1 Acc. (%) | | |
| --- | --- | --- | --- |
|  | RGB | J-6P | SP-8P-1LT |
| E2S-X3D [42] | **98.7** | 10.2 | 10.7 |
| VideoMAE [94] | 90.0 | 9.9 | 9.9 |
| TwoStream-CNN [90] | 97.0 | 15.7 | 10.4 |
| MP(ours) | 96.5 | **69.0** | **49.7** |

| # Trainable Parameters (M) | | | |
| --- | --- | --- | --- |
| ResNet3D | 31.7 | SlowFast | 33.7 |
| I3D | 27.2 | X3D | 3.0 |
| R(2+1)D | 27.3 | MViT | 36.1 |
| MP(ours) | 57.5 | | |

Table S1: **Results of more baselines and our motion perceiver (MP) in BMP tasks.** Our MP demonstrates superior performance than baselines on RGB videos, Joint videos with 6 points (J-6P), and SP videos with 8 points and a lifetime of 1 (SP-8P-1LT). Top-1 accuracy (%) is reported. Best is in bold.

Table S2: **Number of trainable parameters in million (M) for baselines and our MP.** Although our model is larger than the baselines, its superior performance is not a result of the larger number of parameters.

In each trial, participants are shown a video randomly selected from the BMP dataset and are then asked to perform a forced 1-out-of-10 choice test to identify the action in the video. Videos from all conditions are randomly selected and presented in random order. The stimuli are uniformly distributed across BMP conditions, with no long-tailed distribution. These action classes are commonly performed in our daily life and free from cultural bias, as psychophysics experiments show humans can recognize them with nearly 100% accuracy on RGB videos. Almost all subjects are from the US, and we did not collect demographic data. All the experiments are conducted with the subjects' informed consent and according to protocols approved by the Institutional Review Board of our institution. Each subject was compensated.

To control data quality, two checks are implemented in the experiment: (1) Four pre-selected videos in the RGB condition are used as the dummy trials, with two videos representing the "sit down" action class and the other two representing the "stand up" action class. Since these four videos are meant for quality controls. There is no ambiguity in classifying the action classes in these four videos. These four dummy trials are randomly dispersed with the rest of the actual trials in one experiment. Participants who fail to recognize correct actions in any of the four videos are excluded for data analysis. 88 out of the total 90 participants passed the tests in the dummy trials. (2) Each participant is allowed to participate in the experiment only once. All the trials in one experiment are always unique. This is to prevent the subjects from memorizing the answers from the repeated trials.

# E  More Baselines Comparisons

Besides the comparison of State-of-the-Art methods discussed in **Sec. 3.2**, this section offers comparisons with three additional baseline methods. **E2-S-X3D [42]** is a two-stream architecture processing optical flow and spatial information from RGB frames separately. **VideoMAE [94]**, a recent baseline trained on Kinetics in a self-supervised learning manner. **TwoStream-CNN [90]** incorporates the spatial networks trained from static frames and temporal networks learned from multi-frame optical flow. The results in **Appendix, Tab. S1** indicate that our method significantly surpasses all baselines under the J-6P and SP-8P-1LT conditions, highlighting the superior generalization capability of our MP model on the BMP dataset.

Moreover, as stated in **Sec. 3.2**, we explore the upper bound performance of our MP on Joint and SP videos. First, we directly train our MP model on J-6P and test it on J-6P. Its accuracy on J-6P is 95% whereas human performance is 86%. Surprisingly, we found this model trained on J-6P also achieves 55% accuracy in RGB and 71% in SP-8P-1LT, which are far above chance. This implies that our model has generalization ability across multiple modalities. Second, we also train our model on SP-8P-1LT and test it on all three modalities: 43% in RGB, 69% in J-6P, and 93% in SP-8P-1LT. The reasonings and conclusions stay the same as the model directly trained on J-6P. Note that although our model achieves very high accuracy on the in-domain test set (train on J-6P, test on J-6P and train on SP-8P-1LT, test on SP-8P-1LT), its overall performance over all three modalities (RGB, J-6P, and SP-8P-1LT) is still lower than humans (74% vs 88%). This emphasises the importance of studying model generalisation in BMP. There is still a performance gap between AI models and humans in BMP.

## F   Enhanced Motion Perceiver (En-MP)

It is worth noting that our current MP model (see **Sec. 2**) is applied solely to the feature maps from the final attention block of DINO (block 12). We have observed that our MP model can also be effectively applied to early and intermediate-level blocks of DINO. The final prediction is then generated by fusing features across these three blocks (blocks 1, 7, and 12). We refer to this improved model as the **Enhanced Motion Perceiver (En-MP)**.

## G   Viewpoint Discrepancy

As mentioned in **Sec. 4.1**, human behaviors in our study reveal a divergent trend compared to the human performance pattern described in [46]. We point out two reasons: first, the tasks differ between our study and [46]. Our study focuses on action recognition while the work [46] focuses on walking pattern discrimination. Second, we found the performance variations across different camera views depend on the action classes. As shown in **Appendix, Fig. S2**, humans exhibit the best performance when viewing "arm circles" actions from the frontal view, because the movement of the two arms will not overlap in the frontal view. For the videos from the action class "point to something", the 90°view is the best viewpoint since it results in longer movement trajectories of the arm than other viewpoints. Therefore, although humans can achieve the best average accuracy on the 45°viewpoint in our experiment, it does not imply that humans always perform the best on the 45°viewpoint in all action classes.

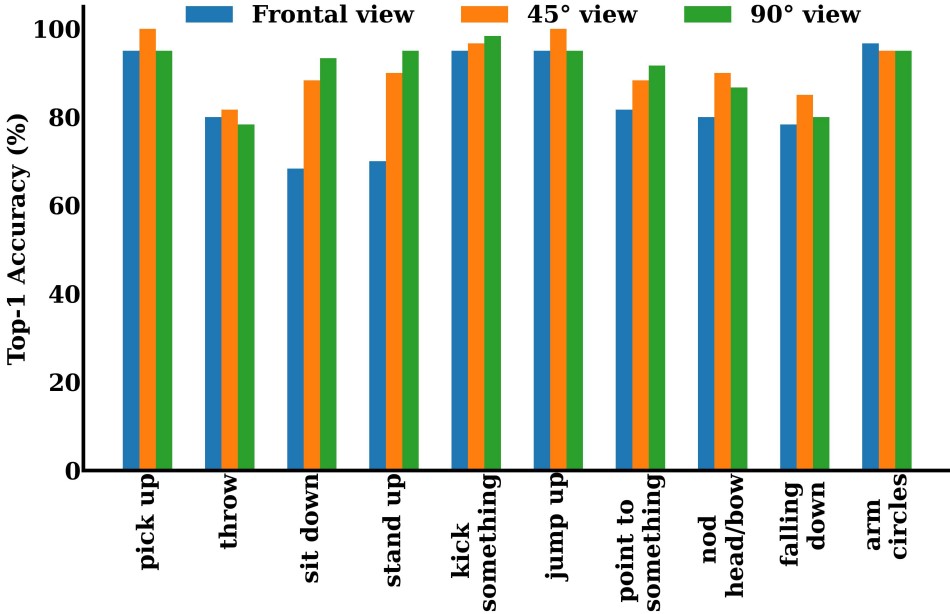

Figure S2: **Human performance across camera views within different action classes on J-6P videos in BMP dataset.** The labels on the x-axis are the 10 action classes in the BMP dataset (**Appendix, Sec. A.2**).

## H   Alignment Between Models and Humans

The alignment between models and humans can be assessed in three aspects: (1) The correlation coefficients between all the AI models and human performance across all the BMP conditions. The results in **Fig. 6** show that our MP has the highest correlation with human performance compared with baselines. (2) The absolute accuracy in all action classes across all BMP properties, which is reported in **Appendix, Fig. S3**. We averaged the accuracies within the same property. It is observed that all AI models demonstrated lower accuracy than humans. However, among all AI models, our MP and En-MP outperform the rest. For example, despite explicitly modelling both motion and visual

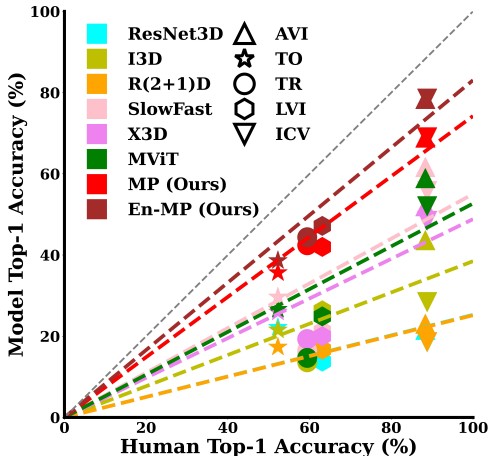
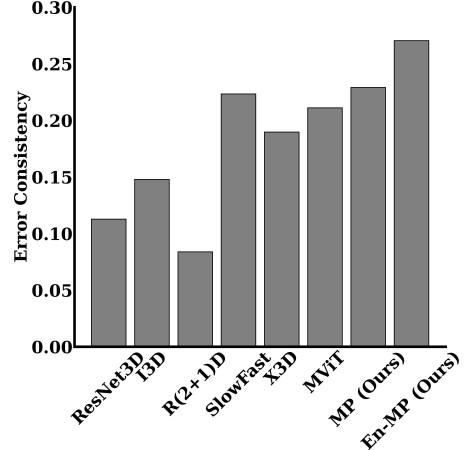

Figure S3: **Our MP and En-MP significantly surpass all existing models on both Joint and SP videos.** A correlation plot between model and human performances is provided, with accuracy averaged over conditions within each property on the same type of stimulus (**Sec. 3.1**). The black dash diagonal serves as a reference.

Figure S4: **Our MP and En-MP show higher error consistency with human performance compared to all existing models.** The Error Pattern Consistency [31] between each AI model and human performance across all BMP conditions is reported.

information in two separate streams, the SlowFast model still falls short in temporal orders. Similarly, MViT, which utilizes transformer architectures, fails to generalize to various temporal resolutions. See **Appendix, Tab. S6** for detailed comparisons. (3) The error pattern consistency between AI models and humans using the metric introduced in [31]. The results in **Appendix, Fig. S4** reveal that the error patterns from our MP and En-MP are more consistent with human error patterns than all the baselines at the trial level.

# I  Baselines with Optical Flow as Input

As mentioned in **Sec. 4.2**, the MP model explicitly uses patch-level optical flow to extract temporal information for classification, whereas the baselines do not. Thus, here we introduce augmented baselines where they are trained on pixel-level optical flows. Specifically, we use GMFlow [105, 106] to extract the pixel-level optical flow from RGB videos in the BMP dataset, and then convert them into three-channel videos based on the Middlebury color code [5]. Taking these optical flow videos in the training set as the input to baselines, the performances on optical flow videos in the testing set and J-6P are shown in **Appendix, Tab. S3**. It is evident that all baselines achieve high accuracy on optical flow videos in the testing set, indicating that they effectively learn to recognize actions from optical flow inputs. However, their best performances on J-6P are significantly worse compared to the MP model (30.83% vs 69.00% in **Appendix, Tab. S6**), suggesting that our MP learns more generalizable motion representations from patch-level optical flows.

|  | ResNet3D [38] | I3D [11] | R(2+1)D [96] | SlowFast [29] | X3D [28] | MViT [25] |
|---|---|---|---|---|---|---|
| OF | **99.23** | 99.02 | 98.67 | 99.12 | 98.35 | 99.19 |
| J-6P | 9.94 | 9.87 | 9.38 | 9.90 | 9.87 | **30.83** |

Table S3: **Results of baselines with optical flow as input.** Top-1 accuracy is reported on optical flow videos (OF) and Joint videos with 6 points (J-6P). Best is in bold.

# J  Visualization of Patch-Level Optical Flow

As outlined in **Sec. 4.2**, we provide a visualization of patch-level optical flow in vector field plots across example video frames for "stand up" action in **Appendix, Fig. S5**. We can see that patch-level

optical flow mostly happens in moving objects (the person performing the action) and captures high-level semantic features. Hence, they are more robust to perturbations in the pixel levels and more compute-efficient.

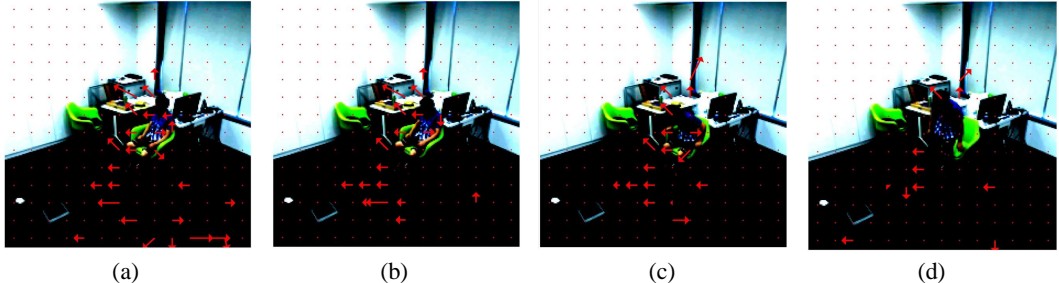

|  | (a) | (b) | (c) | (d) |

Figure S5: **Visualization of patch-level optical flow in vector field plots across a sequence of example video frames from "stand up" action class in the BMP dataset.** Optical flow from (a) Frame 1; (b) Frame 8; (c) Frame 16; (d) Frame 31 to Frame 32.

## K    More Ablation Studies

To investigate the impact of additional hyperparameters and key components in our model, we have conducted further ablation studies. Note that we use the same set of hyper-parameters for our model across all the datasets.

| Top-1 Acc. (%) | $K$ | | | $\alpha$ | | $\tau$ | | $\mu$ | | Full Model |
|---|---|---|---|---|---|---|---|---|---|---|
|  | 5 | 8 | 10 | 1 | 100 | 0.10 | 0.01 | 0.02 | 0.20 |  |
| RGB | 95.40 | 96.24 | 96.14 | 96.66 | 96.21 | **98.17** | 96.84 | 96.77 | 96.35 | 96.45 |
| J-6P | 67.77 | 68.96 | 68.15 | 66.92 | 66.57 | 40.27 | 22.19 | 68.29 | 67.63 | **69.00** |
| SP-8P-1LT | 45.79 | 48.77 | 46.35 | 47.51 | 45.08 | 26.44 | 18.96 | 47.09 | 46.70 | **49.68** |

Table S4: **Ablation of key hyper-parameters in our model.** Top-1 accuracy is reported on RGB videos, Joint videos with 6 points (J-6P), and SP videos with 8 points and a lifetime of 1 (SP-8P-1LT). From left to right, the ablated hyper-parameters are: the number of slots $K$ (**Sec. 2.2**) the weight $\alpha$ of $L_{slot}$ (**Sec. 2.4**), temperature $\tau$ in the patch-level optical flow (**Sec. 2.1**) and temperature $\mu$ in $L_{slot}$ (**Appendix, Sec. B.2**). Best is in bold.

**Temperature $\tau$ in the patch-level optical flow: Appendix, Tab. S4** demonstrates that higher temperature $\tau$ used for computing patch-level optical flows hurt performances, compared with our full model where $\tau = 0.001$. This implies that temperature $\tau$ controls the smoothness of the sequence of flows and lower temperature $\tau$ is beneficial for clustering flows with similar movement patterns.

**Number of slots K:** As shown in **Appendix, Tab. S4**, there is a non-monotonic performance trend versus the number of slots. The number of slots controls the diversity of temporal information captured from the path-level optical flow. An increase in the number of temporal slots can lead to redundant temporal clues, while a decrease might result in a lack of sufficient temporal clues.

**The weight $\alpha$ of $L_{slot}$:** The parameter $\alpha$ regulates the importance of $L_{slot}$ in comparison to other loss functions. A small value of $\alpha$ can impede the diversity in temporal slots, while a large $\alpha$ might adversely affect the optimization of features extracted from flow snapshot neurons and motion invariant neurons. From **Appendix, Tab. S4**, it can be seen that when $\alpha$ is either small ($\alpha = 1$) or large ($\alpha = 100$), there will be a slight degradation in performance compared with our full model with $\alpha = 10$.

**Temperature $\mu$ in $L_{slot}$:** The temperature $\mu$ controls the sharpness of distribution over slots. High temperatures will negatively impact the convergence of slot feature extraction, while low temperatures can produce excessively sharp distributions, potentially impairing the stability of the optimization process during training. **Appendix, Tab. S4** indicates that configuring $\mu$ at either 0.02 or 0.20 are non-optimized options compared with our full model when $\mu$ is 0.05.

| Ablation ID | Time Emb. | $L_{slot}$ | $1_{st}$ Frame Ref. | Top -1 Acc. (%) | | |
|---|---|---|---|---|---|---|
| | | | | RGB | J-6P | SP-8P-1LT |
| 1 | ✗ | ✓ | ✗ | 95.19 | 68.12 | 47.12 |
| 2 | ✓ | ✗ | ✗ | **96.52** | 67.84 | 49.12 |
| 3 | ✓ | ✓ | ✓ | 91.40 | 58.60 | 47.16 |
| Full Model | ✓ | ✓ | ✗ | 96.45 | **69.00** | **49.68** |

Table S5: **Ablation of the time embedding (Sec. 2.4), the loss term $L_{slot}$ (Sec. 2.2) and the referenced frame for calculating path-level optical flow (Sec. 2.1) in our model.** Top-1 accuracy is reported on RGB videos, Joint videos with 6 points (J-6P), and SP videos with 8 points and a lifetime of 1 (SP-8P-1LT). Best is in bold.

**Model components:** The ablation of the time embedding (**Sec. 2.4**) and the loss term $L_{slot}$ (**Sec. 2.2**) in our model is shown in **Appendix, Tab. S5**. As illustrated in **Sec. 2.4**, time embeddings are appended to the activations of flow snapshot neurons. Aligning with [66], time embeddings provide useful temporal context (**A1**). The loss $L_{slot}$ on contrastive walks encourages the slots to capture diverse prototypical flow patterns. Consistent with [102], removal of $L_{slot}$ (**A2**) leads to degraded generalisation performance, indicating the importance of contrastive walk loss between slots and optical flows. Furthermore, if our MP relies solely on the first frame as the reference for computing optical flow (**A3**), the performance will degrade significantly. This is because optical flows are estimated by computing the similarity between feature maps from video frames. The errors in feature similarity matching might be carried over in computing optical flows. Using multiple frames as references for computing optical flows eliminates such errors.

**Downscale pixel-level flows:** We also introduce an MP variation using pixel-level optical flow downscaled to the size of patch-level optical flow as input. Our MP model outperforms this model variation: 96.5% vs 68.8% in RGB, 69.0% vs 12.6% in J-6P, and 49.7% vs 9.4% in SP-8P-1LT. Hence, this implies that DINO captures semantic features that are more effective and robust for optical flow calculation than downscaled pixel-level flows.

**Feature Extractor DINO with ResNet:**In our MP, using ViT-based DINO has demonstrated its effectiveness in biological motion perception. We then conducted an experiment where ViT was replaced with the classical 2D convolutional neural network (2D-CNN) ResNet50, pre-trained on ImageNet, as a feature extractor for video frames. The results show that while the performance of our MP with ResNet50 is lower than with ViT, it still exceeds chance levels. Specifically, the accuracy of DINO-ViT (ours) versus DINO-ResNet50 is: 96.45% vs 80.37% in RGB, 69.00% vs 40.34% in J-6P, and 49.68% vs 40.03% in SP-8P-1LT. Additionally, it outperforms baselines using 3D-CNN backbones, such as ResNet3D, I3D, and R(2+1)D. This suggests that our MP effectively generalizes in BMP, regardless of the feature extraction backbone used.

**Number of Trainable Parameters** As mentioned in **Sec. 4.3**, we present the number of trainable parameters for all baselines and our MP model in **Appendix, Tab. S2**. While our model is larger than the baselines, its superior performance is not attributed to its size. To validate this, We included a SlowFast-ResNet101 variant with 61.9M parameters, which underperforms compared to our model. Specifically, the performance of our model versus SlowFast-ResNet101 is 96.5% vs 99.3% in RGB, 69.0% vs 39.4% in J-6P and 49.7% vs 12.6% in SP-8P-1LT. In addition, we list the number of trainable parameters in millions (M) for each model part of our MP: Flow Snapshot Neuron (0.07M), Motion Invariant Neuron (0M) and Feature Fusion (57.5M). Compared to DINO with 85.8M parameters for image processing, our MP model, appended to DINO, only requires slightly more than half of its size. Yet, it leverages DINO features from static images to generalize to recognize actions from a sequence of video frames.

**Key Frame Analysis** As indicated in **Sec. 4.3**, to explore how the start, end, or development of the action would influence action recognition, we analyze the effect of which frames are essential for the "pick up" action class in one example video. Briefly, we randomly selected X frames among 32 frames, duplicated the remaining frames to replace these selected frames, and observed the accuracy drops, where X = [1,8,16,24,28,31]. When multiple frames are replaced, the performance drop implies the importance of the development of these frames. In total, we performed 1000 times of

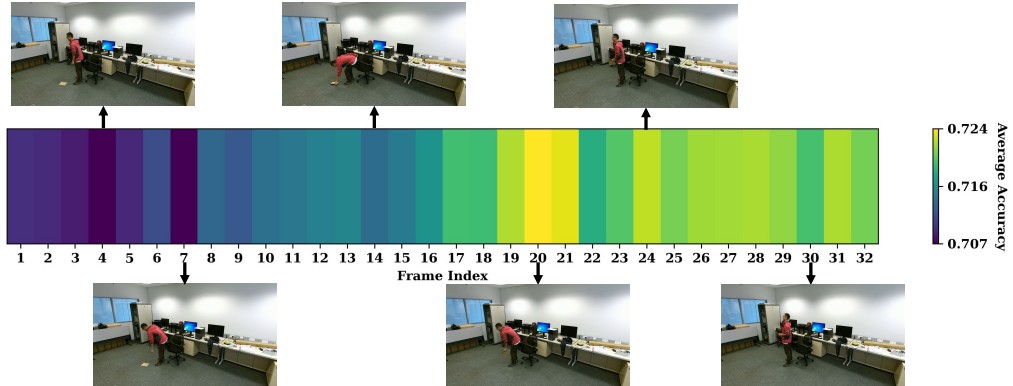

Figure S6: **Frame analysis for the "pick up" action class in one example video from the BMP dataset.** The color in the heat map indicates the video frames which would impact the action recognition accuracy of the model the most. Blue colors represent higher importance.

random frame selections per X and presented the visualization of frame importance by averaging all the accuracy drops over all the random frame selections. The visualization results in **Appendix, Fig. S6** suggest that the fourth and seventh frames are essential for the pick-up class recognition.

## L    Detailed Results of All BMP Conditions

The detailed results of all BMP conditions are shown in **Appendix, Tab. S6**. It is evident that, in comparison to baselines, our motion perceiver (MP) not only delivers enhanced performance but also more accurately mirrors human performance in most BMP conditions. Especially in the case of SP-8P-1LT, our model significantly outperforms the top baseline by a substantial margin of 29.39%. These findings emphasize that our MP model is more effective at recognizing biological motion, demonstrating greater alignment with human behavioral data than existing AI models.

| | ResNet3D [38] | I3D [11] | R(2+1)D [96] | SlowFast [29] | X3D [28] | MViT [25] | MP(ours) | Human |
|---|---|---|---|---|---|---|---|---|
| RGB | 98.56 | 98.74 | **99.02** | 98.63 | 98.67 | **99.02** | 96.45 | 98.00 |
| RGB-R | 70.51 | 68.82 | **76.23** | 62.18 | 66.50 | 68.19 | 62.32 | 61.83 |
| RGB-S | 53.12 | 41.57 | 38.76 | 22.68 | 41.50 | 51.65 | 61.34 | **76.00** |
| RGB-4F | 68.89 | 76.62 | 77.91 | 60.11 | 75.53 | 32.44 | 68.82 | **88.00** |
| RGB-3F | 61.24 | 62.89 | 64.61 | 65.98 | 64.92 | 31.88 | 68.43 | **84.33** |
| J-26P | 27.95 | 75.35 | 33.39 | 77.98 | 67.21 | 76.90 | 72.16 | **92.50** |
| J-18P | 26.12 | **76.51** | 34.66 | 75.74 | 67.80 | 75.46 | 73.17 | / |
| J-14P | 25.77 | 68.15 | 35.22 | 72.30 | 66.68 | 74.61 | 70.37 | / |
| J-10P | 20.75 | 56.04 | 26.90 | 70.54 | 61.03 | 68.05 | **71.73** | / |
| J-6P | 18.54 | 28.41 | 18.57 | 55.72 | 48.38 | 52.00 | 69.00 | **86.33** |
| J-5P | 18.19 | 27.14 | 17.17 | 51.23 | 40.41 | 47.86 | 65.52 | **86.17** |
| J-6P-R | 17.70 | 23.53 | 16.26 | 35.04 | 31.00 | 32.37 | 38.69 | **56.67** |
| J-6P-S | 26.69 | 19.56 | 18.43 | 24.30 | 20.93 | 20.89 | 32.65 | **47.83** |
| J-6P-4F | 16.19 | 10.96 | 15.80 | 21.70 | 20.40 | 12.29 | 41.12 | **63.17** |
| J-6P-3F | 16.50 | 16.22 | 13.83 | 11.27 | 18.22 | 17.03 | 43.71 | **55.67** |
| J-6P-0V | 19.83 | 31.36 | 17.86 | 59.50 | 51.30 | 52.23 | 70.20 | **84.00** |
| J-6P-45V | 17.98 | 30.46 | 18.73 | 58.88 | 53.18 | 55.11 | 69.64 | **91.50** |
| J-6P-90V | 17.78 | 23.43 | 19.14 | 48.85 | 40.79 | 48.74 | 67.15 | **90.83** |
| SP-4P-1LT | 12.96 | 21.91 | 10.50 | 19.10 | 14.12 | 10.64 | 32.76 | **56.33** |
| SP-4P-2LT | 10.57 | 22.02 | 15.06 | 26.09 | 17.77 | 18.40 | 35.01 | **45.36** |
| SP-4P-4LT | 12.64 | 23.07 | 19.31 | 19.77 | 19.59 | 27.53 | 34.76 | **52.50** |
| SP-8P-1LT | 15.59 | 18.43 | 10.53 | 20.29 | 19.91 | 15.06 | 49.68 | **80.33** |
| SP-8P-2LT | 13.45 | 33.85 | 17.49 | 21.00 | 22.54 | 31.07 | 49.33 | **69.64** |
| SP-8P-4LT | 17.35 | 38.97 | 27.39 | 21.31 | 26.23 | 46.63 | 50.49 | **74.47** |

Table S6: **Detailed results of baselines, motion perceiver (MP) and human in all BMP conditions.** Top-1 accuracy (%) is reported. There are 24 conditions in total. Best is in bold. The second best is underlined.

# M  Social Impact

The Motion Perceiver, with its human-like ability to perceive biological motion, brings both promising advancements and concerning implications for society. In the sports and fitness field, it provides valuable tools for improving athletic performance, injury prevention, and personalized training. However, the same capabilities may raise privacy concerns, as the technology could potentially be misused for unauthorized surveillance or tracking of individuals' movements in public spaces. This invasion of privacy might extend to analyzing behavior patterns and gait. Furthermore, there is a risk of discrimination if the method is employed in contexts like employment screening or law enforcement profiling, where certain movement patterns might be unfairly associated with specific demographics or lead to biased judgments.

