# OpenReview forum: "Flow Snapshot Neurons in Action: Deep Neural Networks Generalize to Biological Motion Perception"
_NeurIPS.cc/2024/Conference — NeurIPS 2024 poster_

### Official Review · Reviewer_eynA · 2024-07-08

**Soundness:** 3
**Presentation:** 3
**Contribution:** 3
**Rating:** 7
**Confidence:** 4

**Summary:**

This paper uses AI models to investigate how humans can accurately predict actions based solely on minimal action patterns without prior training. The authors propose a new motion recognition model called Motion Perceiver (MP). This new model first extracts patch-level semantic features from images using a DINO-pretrained ViT model, then computes patch-level optical flow from these features, and develops two pathways featuring motion-invariant representations and slot-based representations from these optical flows. They show that this new model yields more human-like generalization performance on recognizing actions from videos of joints and sequential positions than existing AI models. Their new model also shows some human-like behavior patterns when performing action recognition. They further present ablation studies showing which components in this new model are critical.

**Strengths:**

This paper contains solid work on collecting/constructing training and testing datasets, developing a new computational model, training existing baselines on the datasets, comparing the new model to these existing baselines, and showing the influence of different architectural designs in the new model. The authors have also collected human psychophysics data through MTurk on their minimal motion stimuli, which is useful for the community.

The new model proposed by the authors outperforms the existing baselines by significant and sometimes large margins. Both the new model and baselines are evaluated on various scenarios to test their generalizability performance.

The paper is well written, with a lot of details in the appendix and clear descriptions in the main text.

**Weaknesses:**

1.	This paper needs stronger baselines. The authors train their baselines on the small amount of natural RGB videos collected by them. However, humans perceive a lot of videos during their development in the real world. This means a stronger and more human-like baseline is finetuning from a pretrained video model using self-supervised learning algorithms such as VideoMAE, instead of starting from random initializations. Moreover, the fact that their new model (MP) uses a DINO-pretrained visual encoder makes the existing baselines even weaker, since all the baselines are trained from scratch on the small amount of videos. Fixing this would make the comparison much more reasonable and fairer.

2.	The authors also need to show how good models directly trained on these joints or sequential points videos can perform on this task. It is important to get a sense about how far away this new model is compared to models directly trained on within-domain videos. This can also help tell whether human is better than these models or not.

3.	Suppose this work aims to get a more human-like model through this human-like learning curriculum compared to just training a model on the joint or sequential point videos. In that case, the authors need to show whether their new model is quantitatively more human-like compared to the other models. This can include explaining per-category action-recognition performance or the error patterns when recognizing actions.

I appreciate the authors' responses to these points. The first point is well explained. Please also remember to include this training detail of the baseline models in the final version. The additional experiments on the second point are also convincing. I raise my score from 6 to 7.

**Questions:**

See the weakness points.

**Limitations:**

The authors discussed the limitations of the paper.

---

> ### Author Rebuttal · Authors · 2024-08-07
>
> **[eynA.1 - Unfair comparison with baselines and VideoMAE]** All the baselines are action recognition models pre-trained on Kinetics 400. In contrast, the DINO architecture in our MP is pre-trained on ImageNet. All the models are then fine-tuned on RGB videos in the training set of our BMP dataset. We would like to emphasize that it is remarkable that our MP can still outperform all the baselines even though it has never been pre-trained on any video datasets.
>
> As requested by the reviewer, we now include VideoMAE [b] for comparison with our MP. The model is trained on Kinectics with a self-supervised learning method. Just like other baselines, the performance of VideoMAE is still inferior to our model as shown in Tab R1. We will include this experiment in the final version.
>
> **[eynA.2 - Train on joint or SP videos]**  That is a great suggestion! We now directly train our model on J-6P and test on J-6P. Its accuracy on J-6P is 95.4% whereas human performance is 86%. Surprisingly, we found this model trained on J-6P also achieves 55% accuracy in RGB and 71% in SP-8P-1LT, which are far above chance. This implies that our model has generalization ability across multiple modalities. Similarly, we also train our model on SP-8P-1LT and test it on all three modalities: 42.7% in RGB, 69.3% in J-6P, and 93% in SP-8P-1LT. The reasonings and conclusions stay the same as the model directly trained on J-6P.
>
> Note that although our model achieves very high accuracy on the in-domain test set (train on J-6P, test on J-6P and train on SP-8P-1LT, test on SP-8P-1LT), its overall performance over all three modalities (RGB, J-6P, and SP-8P-1LT) is still lower than humans (73.6% vs 88.2%). This emphasises the importance of studying model generalisation in BMP. There is still a performance gap between AI models and humans in BMP.
>
> **[eynA.3 - human-model alignment]**  The reviewer asked us to compare the human-model alignment between our model and models trained directly on J-6P or SP-8P-1LT. However, this would defeat the purpose of our study. We argue that the generalization ability of a model should arise from well-designed architectures, visual inputs, and learning rules. Augmenting data to tackle the generalisation problem is one engineering and effective approach in computer vision [d-h]. However, just like how humans have been exposed to only RGB naturalistic videos and meanwhile can generalize to multiple types of BMP stimuli, we are interested in developing AI models capable of generalizing to BMP like the way humans do without any data augmentations on the visual inputs during training.
>
> The alignment between models and humans can be assessed in three aspects: (1) the absolute accuracy of a model matches with human accuracy in all the action classes across all BMP conditions, (2) the relative change in accuracy across all BMP conditions is consistent in terms of correlation values, (3) the error pattern consistency between models and humans is high. We present the results in these three aspects for our model and all the baselines.
>
> Similar to Fig 6, we now present Fig R3 in the rebuttal PDF where each dot on the scatter plot indicates the absolute accuracy of a model for every action class for all BMP conditions. If a model and humans achieve the same accuracy in the same action class across all five fundamental properties in BMP, the slope of the fitted linear line is 1 and all the markers would be at the diagonal line. The slope of close to 1 in our model in Fig R3 suggests that our model performs competitively well as humans in all action classes in BMP.
>
> To evaluate the relative change in accuracy across all BMP conditions, we now compute the correlation values for all the models and present the results in Tab R5. Our MP shows the highest correlation with human performance compared with all the baselines.
>
> Lastly, we use the error consistency metric introduced in [p] to evaluate all models’ error consistency with humans and report the results in Tab R6. It turns out that our model achieves the highest error consistency with humans. This implies that humans and our model tend to make similar mistakes at the trial levels.
>
> Overall, the absolute accuracy in Fig R3, the correlation value in Tab R5, and the error consistency score in Tab R6 suggest that our model aligns with humans much more than all the baselines.

---

### Official Review · Reviewer_WFYf · 2024-07-12

**Soundness:** 3
**Presentation:** 3
**Contribution:** 3
**Rating:** 6
**Confidence:** 4

**Summary:**

The paper introduces a novel, neuroscience-inspired approach to performing biological motion perception from videos. The videos are part of a dataset that the authors also introduce, depicting 10 different actions. The videos range from fully RGB frames to point-light displays that only cover the joints. While humans have been very successful at solving such tasks, even with limited data, AI models struggle. For this reason, the authors propose an elegant approach that involves obtaining optical flow from feature maps of an ImageNet-pretrained network and creating a system of slots to organize important features for the task. Additionally, they introduce features that account for scale and invariance. Overall, the method shows significant improvement over the tested baselines.

**Strengths:**

The paper introduces a novel dataset designed to advance the field of action recognition in videos. This benchmark dataset challenges most of the baseline methods in the field. Additionally, the paper presents an elegant approach to combining features over time, inspired by loosely but interestingly applied neuroscience insights. This approach appears to effectively motivate a method that performs better under almost all conditions of the proposed dataset.

**Weaknesses:**

My main question/concern is whether the baselines are fairly compared with the proposed method. It might be worth adding learning curves to indicate whether the models have effectively converged. Additionally, including a parameter count to express whether the method is more or less computationally expensive than the baselines would provide useful insight.

I might have missed it but it seems that the inspiration to take  ViT as a feature extraction is not well motivated. Perhaps worth mentioning if there is some rationale that  harmonizes with neuroscience or if its a engineering choice.

Another question/concern is that the connection back to neuroscience seems weak. However, there might be a potential opportunity arising from this approach. The ablation studies suggest a certain dependency on time resolution. It might be beneficial to show which frame in the sequence is the most important and how the start, end, or development of the action is crucial for action detection. This could tie back to neuroscience and provide testable hypotheses, thereby strengthening the paper.

**Questions:**

I add them in the weakness section.

**Limitations:**

yes

---

> ### Author Rebuttal · Authors · 2024-08-07
>
> **[WFYf.1 - Fair comparison and learning curves]** All the baselines are action recognition models pre-trained on Kinetics 400. In contrast, the DINO architecture in our MP is pre-trained on ImageNet. All the models are then fine-tuned on RGB videos in the training set of our BMP dataset. We would like to emphasize that it is remarkable that our MP can still outperform all the baselines even though it has never been pre-trained on any video datasets.
>
> As requested by the reviewer, we present the learning curves of our MP and MVIT: the loss of our MP converges to near zero after around 7000 iterations with a top-1 accuracy of 97% in the validation set. Similarly, the loss of MVIT converges to near zero after around 1250 iterations with top-1 accuracy of around 97% as well. We note that MViT converges faster than our MP due to the difference in the pre-trained datasets. Both models achieve near-perfect performance in RGB videos; however, our model significantly surpasses MViT in joint videos and SP videos as shown in Table S3 in Appendix F. This emphasizes that our MP is better at generalization in BMP.
>
> **[WFYf.2 - Parameter count]** As also requested by the reviewer, we now add Tab R2 listing the number of trainable parameters for all the models. Note that our model is larger than the baselines in the paper. However, we argue that the superior performance of our model is not a result of the larger number of parameters. To demonstrate this, we included a variation of SlowFast-ResNet101 containing 61.9M parameters. Despite its large size, its generalisation performance in BMP is still inferior to ours. The performance of ours versus SlowFast-ResNet101 are: 96.45% vs 99.26% in RGB, 69.00% vs 39.43% in J-6P, and 49.68% vs 12.64% in SP-8P-1LT.
>
> **[WFYf.3 - Motivation of ViT]** The reviewer is correct. We used ViT because it is one of the state-of-the-art architectures for image processing with the best performances in several computer vision tasks. It is purely an engineering choice.
>
> Following up on the reviewer’s advice, we also conducted an experiment where we replaced ViT with the classical 2D-convolutional neural network (2D-CNN) ResNet50 pre-trained on ImageNet as a feature extractor from video frames. Evidence in neuroscience [l-o] has suggested that 2D-CNN models are bio-plausible models capable of predicting behaviours and neural responses in primates. Results show that the performance of our MP with ResNet50 is lower than our original model with ViT but still higher above chance. The accuracy of DINO-ViT (ours) versus DINO-ResNet50 are: 96.45% vs 80.37% in RGB, 69.00% vs 40.34% in J-6P, and 49.68% vs 40.03% in SP-8P-1LT. Moreover, it outperforms the baselines with 3D-CNN as backbones, such as ResNet3D, I3D and R(2+1)D. This suggests that our MP is effective at generalization in BMP regardless of its feature extraction backbones.
>
>
> **[WFYf.4 - Connection to neuroscience]** The stimulus design in our paper is inspired by neuroscience works. A non-exclusive list of neuroscience papers using point-light displays and studying different BMP conditions are [30,82,9,46,6,16,34,40,64,69,89] in the References. In our work, we also introduced a computational model for BMP. We did not make any claims that our model is biologically plausible as we agree with the reviewer that the model is only loosely inspired by neuroscience.
>
> The reviewer also raises this interesting question on how the start, end, or development of the action would influence action recognition. To address this question, we analyze the effect of which frames are essential for the pick-up action class in one example video. Note that we are unable to systematically and rigorously test all action classes due to the limited time of the rebuttal.
>
> Briefly, we randomly selected X frames among 32 frames, duplicated the remaining frames to replace these selected frames, and observed the accuracy drops, where X = [1,8,16,24,28,31]. When multiple frames are replaced, the performance drop implies the importance of the development of these frames. In total, we performed 1000 times of random frame selections per X and presented the visualization of frame importance by averaging all the accuracy drops over all the random frame selections. See Fig R1 in the rebuttal PDF. The visualization results suggest that the fourth and seventh frames are essential for the pick-up class recognition.
>
> In addition to the question raised by the reviewer, our approach also raises other intriguing neuroscience questions, such as what are the neural basis for motion-invariant neurons that are crucial when video frames are shuffled or reversed. Our work takes an initial step in connecting artificial and biological intelligence in BMP. These bio-inspired architectures can help validate certain hypotheses in neuroscience, and insights from neuroscience can inform the design of better AI systems. We will include these discussion points in the final version.

---

> > ### Comment · Reviewer_WFYf · 2024-08-11
> >
> > I want to thank  the authors for the new experiments provided, including the selection of frame one that I suggested. I increase one point my score.

---

> > > ### Author Response · Authors · 2024-08-11
> > >
> > > We sincerely appreciate the time and effort you have put into thoughtfully reviewing our paper. Your feedback and guidance have been helpful in enhancing our work. Thank you again for your valuable comments and support!

---

### Official Review · Reviewer_buWx · 2024-07-13

**Soundness:** 3
**Presentation:** 3
**Contribution:** 2
**Rating:** 5
**Confidence:** 4

**Summary:**

The paper introduces the Motion Perceiver (MP), a novel AI model designed to improve the generalization of action recognition in biological motion perception (BMP) tasks. It leverages patch-level optical flows and introduces flow snapshot neurons that learn and store prototypical motion patterns and motion-invariant neurons that maintain the consistency of motion recognition across temporal sequences. The authors create a comprehensive BMP benchmark dataset with 62,656 video stimuli across 24 BMP conditions and demonstrate that MP surpasses existing models by 29% in top-1 action recognition accuracy. The MP model’s performance closely aligns with human behavior in BMP tasks, providing significant insights into the underlying mechanisms of BMP and paving the way for more generalizable AI models in action recognition.

**Strengths:**

- Originality: The Motion Perceiver (MP) model, which uses patch-level optical flows and novel neuron mechanisms (flow snapshot neurons and motion-invariant neurons), is a creative and original approach to improving action recognition in BMP tasks. The BMP benchmark dataset is also a significant contribution that provides a new standard for evaluating both human and AI performance in BMP tasks.

- Quality: The paper demonstrates a well-designed and rigorous methodology, including detailed descriptions of the MP model architecture, the BMP dataset, and the experimental setup. The extensive evaluation of the MP model against existing AI models and its comparison with human performance provide strong evidence supporting the paper’s claims.

- Clarity: The paper is well-written and clearly explains the technical details of the MP model and the experimental methodology. Figures and tables are effectively used to illustrate the key concepts and results. The paper also situates its contributions within the broader context of existing research in AI and motion perception, providing a clear understanding of the novelty and significance of the work.

**Weaknesses:**

- Limited Contextualization: Although the paper does a good job explaining the MP model and its benefits, it could benefit from a more detailed comparison with prior work. Specifically, a deeper discussion on how the proposed method improves upon existing models in terms of architectural innovations and performance metrics would provide clearer context. Particularly, lines 56-57 state that “In contrast to many existing pixel-level optical flow models [83, 84, 79, 91, 71], ….”, but some of the references are actually representation-level flow models, e.g., [71]. More comprehensive comparisons to those prior works (theoretically and/or empirically) would make the claim more convincing.

- Dataset Limitations and Biases: The paper introduces a comprehensive BMP benchmark dataset, but it would be beneficial to address potential limitations and biases within this dataset. For example, discussing the diversity of the actions, the representation of different demographics, and the potential impact of these factors on the model’s performance would provide a more balanced view of the dataset’s strengths and limitations. Most importantly, the authors designed two scenarios (joint videos and sequential position actor videos), and claim they are good benchmarks for recognizing motion patterns. This claim is not well justified. For example, why not use sketches, optical flows, or other types of motion representation?

- Results Not Supportive Enough: In Fig. 4, it seems that the proposed method only outperforms on J-6P and J-5P, and is inferior on other levels. In this case, it is hard to claim that the proposed method has better generalization. While the J-6P and J-5P are most abstract and challenging, there is no evidence that those two levels are the best evaluators for generalization on action recognition. Additionally, the comparison methods are mostly dated; more recent baselines are needed. There is no mention/control on the efficiency and computation cost as well, making the comparisons less meaningful.

- Contribution: The biggest weakness is the overall contribution of the paper. While the paper is strong in proposing a new neural architecture and a benchmark, the proposed task itself is not justified with theoretical or practical value. There is no valid proof that joint and SP videos are good representations of biological motion (compared to other motion representations). More specifically, the joints are from the skeleton and keypoints of a human body, which limits its generalization to other objects. Regarding human action recognition itself, it is plausible that having a keypoint modality in the training procedure could potentially solve the problem of distribution shift (RGB to abstract motion representation). While the idea is well-motivated, the approach does not strongly support the claim; the methodology, the proposed benchmark, and empirical results will all benefit a lot from further study.

**Questions:**

Most of my questions are described in Weaknesses. Some additional or specific ones:

1. What's the runtime of the proposed method compared to baselines?
2. Have you tried using the pixel-level optical flow downscaled to patch level and compared that to the patch flow?

**Limitations:**

Yes.

---

> ### Author Rebuttal · Authors · 2024-08-07
>
> **[buWx.1-More comparisons to prior works]** Thanks. We will remove [71] from this sentence in the final version. Moreover, we added two more new baselines [a] and [c]. In [a], E2-S-X3D is a two-stream architecture processing optical flow and spatial information from RGB frames separately. In [c], TwoStream CNN incorporates the spatial networks trained from static frames and temporal networks learned from multi-frame optical flow. Results in Tab R1 show that our method outperforms both baselines, demonstrating the superior generalization ability of our MP model on the BMP dataset.
>
> **[buWx.2-Dataset limitation and bias]** We introduced the 10 action classes in Appendix A1.1 and explained our dataset split in Sec 3.1. Stimuli are uniformly distributed across BMP conditions, with no long-tailed distribution in training or testing. These action classes are common and free from cultural bias, as psychophysics experiments show humans can recognize them with nearly 100% accuracy. Almost all subjects are from the US, and we did not collect demographic data. We are open to testing potential biases if specified.
>
> Tab R4 shows human and model performance in per-category accuracy. We observed variations across BMP conditions. For example, both humans and our model have low accuracy for the stand-up class in the TO property, as altering frame order can confuse it with actions like sitting down or falling down.
>
> We benchmarked humans and models using Joint and SP videos, which are established stimuli for studying human motion perception in vision science, psychology, and neuroscience. We followed the designs from notable works [30,82,9,46,6,16,34,40,64,69,89] in the References to leverage their discoveries and compare our findings.
>
> We are unclear about the specific "sketches" the reviewer refers to. Studies [i-k] indicate that current action recognition models often rely on static features.  Point-light displays reduce these confounding factors by minimizing visual information, thereby highlighting the ability to perceive motion. The reviewer also suggested using optical flows as stimuli, but their suitability for studying motion perception is unclear due to potential colour-based confounding.
>
> Finally, we do not claim Joint or SP videos with point-light displays are the only stimuli for studying biological motion perception. Like CIFAR-10's role in object recognition decades ago, our work introduces initial datasets for biological motion perception, encouraging community contributions.
>
> **[buWx.3-Results not supportive enough]** Yes, the reviewer correctly points out our model's performance in Fig 4. However, Fig 4 shows only one aspect of generalization in Joint videos. Tab S3 shows our model beats baselines in 17 of 24 BMP properties, with some gaps as large as 30%.
>
> Our current MP model is applied only to feature maps from DINO's last attention block (block 12). We found it can also be applied across early and middle blocks (e.g., blocks 1, 7 and 12), with final predictions based on feature fusion across these blocks. As shown in Tab R3, it outperforms the second-best MViT baseline across all Joint cases. We will add this enhanced MP model in the final version.
>
> As requested by the reviewer, we added three more baselines. Two are introduced in **[buWx.1 - More comparisons to prior works]**. Additionally, we added VideoMAE [b], a recent baseline trained on Kinetics in a self-supervised learning manner. Our model also significantly outperforms VideoMAE, as shown in Tab R1.
>
> We also added Tab R2, listing the number of trainable parameters for all the models. Our model size is larger than baselines, but its superior performance is not due to size. We included a SlowFast-ResNet101 variation with 61.9M parameters, which performs worse than ours. The performance of ours versus SlowFast-ResNet101 are: 96.5% vs 99.3% in RGB, 69.0% vs 39.4% in J-6P, and 49.7% vs 12.6% in SP-8P-1LT. We also discussed runtime efficiency comparisons for all models. See **[buWx.5-Runtime comparison]** for more details.
>
> **[buWx.4-Theoretical and practical value]** The stimulus design in our BMP experiments is inspired by well-established works in vision science, psychology, and neuroscience. See more details in **[buWx.2-Dataset limitation and bias]**. We do not claim that joint videos and SP videos are the only ways to study the generalization of motion perception. There are definitely other types of stimuli testing the generalization of motion perception for other objects, such as random noise animated with motion extracted from real objects shown in [a].
>
> The reviewer suggested training the models on RGB videos with the keypoint modality, but this would defeat our study's purpose. We argue that a model's generalization ability should arise from well-designed architectures, visual inputs, and learning rules. While data augmentation is an effective engineering approach in computer vision [d-h], we aim to develop AI models that generalize to BMP stimuli as humans do, without augmenting visual inputs during training.
>
> **[buWx.5-Runtime comparison]** All the baselines are pre-trained on Kinetics 400, whereas the DINO architecture in our MP is pre-trained on ImageNet. This difference makes fair runtime comparisons challenging. As expected, our MP has a longer runtime than the baselines due to the pre-trained dataset differences. Nonetheless, it is noteworthy that our MP outperforms all baselines despite not being pre-trained on any video datasets.
>
> **[buWx.6-Downscale pixel-level flows]** As suggested by the reviewer, we conducted an MP variation using pixel-level optical flow downscaled to the size of patch-level optical flow as input. Our MP model outperforms this model variation: 96.5% vs 68.8% in RGB, 69.0% vs 12.6% in J-6P, and 49.7% vs 9.4% in SP-8P-1LT. This implies that DINO captures semantic features that are more effective and robust for optical flow calculation than downscaled pixel levels.

---

> > ### Comment · Reviewer_buWx · 2024-08-09
> >
> > Thanks the authors for the detailed rebuttal. The additional results with new baselines are beneficial. On the contribution side, it is still not convincing to me that the proposed BMP design is a good representative of generalization capability of motion perception, but I think the paper overall (methodology and evaluation framework) does make meaningful contributions to the field towards more generalizable perception. I have increased my score therefore.

---

> > > ### Author Response · Authors · 2024-08-10
> > >
> > > Thank you for your constructive feedback! We are glad that our responses have addressed most of your questions.
> > >
> > > The reviewer is still not convinced that the proposed BMP design is a good representative of the generalization capability of motion perception. We respectfully disagree with this point and provide the following arguments:
> > >
> > > Point-light displays (our BMP designs) are a highly effective tool for testing motion perception generalization because they isolate the motion cues from other visual information, allowing researchers to focus purely on how the brain perceives and interprets movement [30,82,9,46,6,16,34,40,64,69,89]. Here are some key reasons why point-light displays are particularly useful for studying the generalization capability of motion perception:
> > >
> > > **1. Minimalistic Representation**
> > >
> > > Point-light displays strip away all extraneous visual details, such as texture, colour, and form, and represent motion using just a few points of light corresponding to the major joints of a moving figure. This minimalistic representation ensures that any perception of motion is based purely on the movement of these points, allowing researchers to study the fundamental mechanisms of motion perception.
> > >
> > > **2. Focus on Motion Cues**
> > >
> > > Since point-light displays lack detailed structural information, the observer's ability to perceive motion relies solely on dynamic cues. This helps researchers understand how motion information alone contributes to the recognition of objects or actions, without the influence of other visual features.
> > >
> > > **3. Generalization Across Contexts**
> > >
> > > Because point-light displays remove contextual and visual details, they are an excellent way to test whether motion perception can generalize across different contexts. For example, people can still recognize a walking figure from point-light displays even when specific visual details are absent, demonstrating the brain's ability to generalize motion patterns.
> > >
> > > **4. Biological Motion Perception**
> > >
> > > Point-light displays are particularly effective for studying biological motion perception, which is the ability to perceive complex movements like walking, running, or dancing. These displays can show how well the visual system can recognize and interpret the patterns of movement that are characteristic of living beings, even with minimal visual information.
> > >
> > > Overall, the use of point-light displays provides a controlled environment to study the fundamental aspects of motion perception and its generalization across different contexts and conditions.
> > >
> > > Finally, we sincerely appreciate the time and effort you have put into thoughtfully reviewing our paper. Your feedback and guidance have been helpful in enhancing our work. Thank you again for your valuable comments and support!

---

### Official Review · Reviewer_nybp · 2024-07-15

**Soundness:** 3
**Presentation:** 4
**Contribution:** 4
**Rating:** 8
**Confidence:** 4

**Summary:**

This paper proposes a new biologically inspired architecture for action recognition. The motion perceiver computes a patch-level optical flow from DINO features which is then processed in a two-stream architecture with one pathway using slot-attention to recognize different motion patterns and the other one integration over motion to get a time-independent motion signal. The paper also contributes a new dataset which provides multipe point light versions of the videos together with human performance data. The model is trained on RGB videos and evaluated on the point light videos. The authors show that the model's performance is well aligned with human performances on the different stimulus versions and that it outperforms other ML models on the least-information stimuli.

**Strengths:**

* very interesting study linking neuroscience, psychophysics and deep learning
* Interesting modelling architecture with multiple relevant and significant contributions:
  * patch based optical flow on DINO features is a nice idea and seems to work very well
  * slot attention on the optical flow data is also a nice idea
  * motion invariant neurons which integrate over motion patterns
* well-designed dataset
* convincing and interesting results that are well presented and discussed
* interesting ablation study with very good discussion that goes beyond a simple table
* very well written paper with clear structure.

**Weaknesses:**

* I think at least one relevant comparison is missing: Illic et al, "Is Appearance Free Action Recognition Possible" (ECCV 2022, https://f-ilic.github.io/AppearanceFreeActionRecognition) asks a very related question and also uses somewhat similar stimuli (white noise stimuli, but also different kinds of dot stimuli) and also introduce a dataset to that end (AFD). Illic et al claim that their twostream architecture can solve the appearance free stimuli and at least on the homepage they also mention dot stimuli close to the ones used in the present paper. However, so far I couldn't find any dot stimuli results in the actual paper. Beyond this paper, I think there are more two-stream architecture models that might be relevant to include. This is overall the main reason that kept me from increasing the rating of the paper.
* the claim "Our model demonstrates human-like robustness to minimal visual information" (Figure 4) seems to strong for me. For most of the stimulus types, MP is not even the best competitor, only for J-6P and J5-P it outperforms all other models. But even then the performance drop compared to RGB seems to be 3 times more than for humans. The same holds for l282 "There is a slight decrease in accuracy when transitioning from RGB to J-6P for both humans and MP", where humans seem to be around 90% and MP around 70%. Interestingly, Figure 4 suggests that most of the performance drop comes from going from RGB to the point light display, because the drop happens already at J-26P, after which the performans of MP mostly stays constant.
* I'm missing more discussion of when and where "classical AI models fail or succeed and the alignment with humans". I think there is a bit to be learned from Appendix F. Especially, I would find it interesting to compute corellations with human scores across the different stimuli. Figure 3 suggests that MP is well aligned with human BMP. But I would like to know if it is also more aligned than other models (which I think would be a very strong additional result).
* notation sometimes a bit convoluted (M, \hat M, \tilde M, ...). Maybe sometimes it's worth using more descriptive names like "M_\text{invariant}"


**Update**: After the rebuttal, I increased my review from 7 to 8.

**Questions:**

* How many trainable parameters has the final model, and how are they distributed over the model parts?
* given that the DINO based patch-level optical flow seems to be a major pillar of the model, I would love to get a better idea of how it performs on some example videos or images. It could be nice to have some vector field plots in the Appendix or a video in the Supplementary Material
* Caption Figure 1: "AI models are trained to learn to recognize actions": Is this on purpose? I would say they are either trained to recognize actions or they learn to recognize action. Being trained to learn to recognize would imply some meta learning.
* l98 "Considering Ft as a 2D grid of patches in N = H × W where H and W are the height and width of Ft " this sentence seems to be missing a subject
* l127: "This dense optical flow estimation approach at the patch level captures a richer set of motion dynamics, providing a more comprehensive analysis of movements throughout the video" I find it interesting that the patch-level optical flow is defined relative to a fixed reference frame instead of, e.g. always the first frame and then actually uses all frames as reference. Did the authors check how much does this help compared to only using e.g. the first frame? Right now this sentence doesn't seem to have any supporting evidence
* l158: " Every patch-level optical flow in Ô can be projected into four orthogonal motion components along +x, −x, +y and −y axes": I think I'm missing something here, but how are motions in +x and -x orthogonal?
* l163 "we obtain motion invariant matrix": "we obtain THE motion invariant matrix"?
* l270 " Interestingly, shuffling has a lesser impact on human performance compared to reversing RGB videos (RGB-R versus RGB-S)" technically, the two performances might be identical within the margin of error if I'm not mistaken.
* Figure 4: given the size of the error bars in Figure 3, I think it would be nice to also have error bars in this figure. Since it would make the figure harder to read, I think it would be enough to have a larger version in the appendix with error bars that can be referred here.
* l322: I agree that ablations A1 and A2 show that both pathways are important, but they seem to be hardly of similar importance. Removing the motion invariant pathway results in a drop of performance, while removing the flow snapshot neuron pathway essentially results in model performance collapsing. So I would say that the flow snapshot neuron pathway is the crucial part, and invariance helps further.
* l329 "In A4, the model takes patch-level optical flows as inputs and directly utilizes them for feature fusion and action classification without flow snapshot neurons" Unless I'm missreading the table, A4 has flow shapshot neurons?
* l349 "Psychophysics experiments on this dataset were conducted, providing human behavioral data as an upper bound": Why upper bound? I don't see why models couldn't outperform humans in principle.
* references to the appendix could be a bit more clear, e.g. "see Appendix, Sec B" instead of "see Sec B"

**Limitations:**

Limitations are well discussed in the main paper. Societal impact is not discussed because "There are no societal impacts, to the best of our knowledge" (l980), with which I would slightly disagree. There is always potential societal impact from building more human like models (good as well as bad). But this is mostly a nitpicking point that doesn't affect my review of the paper.

---

> ### Author Rebuttal · Authors · 2024-08-07
>
> **[nybp.1 - missing AFD and two-stream models]** Thanks! We will cite and discuss this paper in the final version. Complementary to the AFD dataset in Illic et al., we introduce the BMP dataset containing stimuli on point-light displays, also commonly studied in psychology and neuroscience.
>
> In addition to the two-stream SlowFast baseline already included in our paper, we add one more two-stream network E2-S-X3D in Illic et al. for comparison on our BMP dataset. It underperforms our MP model by a large margin (see Tab R1). This demonstrates the importance of FSN and MIN in our model.
>
> **[nybp.2 - Fig4 and claim in l282]** Yes, the reviewer is correct. Note that our current MP is only applied on the feature maps from the last attention block of DINO (block 12). We found that our MP model can also be applied across early and middle-level blocks of DINO. The final prediction is made based on the feature fusion across these three blocks (blocks 1, 7 and 12), with only a 7.9% accuracy difference on the J-6P compared to humans (see Tab R3). It also outperforms the second-best MViT baseline across all the number of joints. We will add this enhanced MP model in the final version and revise the claims in Fig 4 and l282.
>
> **[nybp.3 - alignment and correlation]** The alignment between models and humans can be assessed in three aspects: (1) The absolute accuracy in all action classes across all BMP conditions, which is reported in Sec 4.2 and Fig 6. The slope of close to 1 in our model in Fig 6 suggests that our model performs competitively well as humans in all five properties in BMP. (2) The correlation values for all the models across all the BMP conditions. The results in Tab R5 show that our MP has the highest correlation with human performance compared with baselines. (3) The error pattern consistency between models and humans using the metric introduced in [p]. The results in Tab R6 indicate that our model achieves the highest error consistency with humans at the trial level.
>
> **[nybp.4 - notations]** We will make the notations clearer in the final version.
>
> **[nybp.5 - model parameters]** We list the number of trainable parameters in million (M) for each model part of our MP: FlowSnapshot Neuron (0.07M), Motion Invariant Neuron (0M), Feature Fusion (57.5M),
> Compared to DINO with 85.8 million (M) parameters for image processing, our MP model, appended to DINO, only requires slightly more than half of its size. Yet, it leverages DINO features from static images to generalize to recognize actions from a sequence of video frames.
>
> **[nybp.6 - visualization of optical flow]** See Fig R2 in the rebuttal PDF for a visualization of patch-level optical flow in vector field plots across example video frames for “stand up” action. We can see that patch-level optical flow mostly happens in moving objects (the person performing the action) and captures high-level semantic features. Hence, they are more robust to perturbations in the pixel levels and more compute-efficient.
>
> **[nybp.7 - minor in language]** Thanks. We will fix it:" AI models are trained to recognize actions".
>
> **[nybp.8 - minor in language]** Thanks. We will fix the grammar mistakes.
>
> **[nybp.9 - first frame as reference frame]** As the review suggested, we add an ablation study where our MP only uses the first frame as the reference frame to compute optical flow. Our MP significantly outperforms this ablated version. Top-1 accuracy for MP and ablated MP are: 96.5% vs 91.4% in RGB; 69% vs 58.7% in J-6P, and 49.7% vs 47.2% in SP-8P-1LT. Optical flows are estimated by computing the similarity between feature maps from video frames. The errors in feature similarity matching might be carried over in computing optical flows. Using multiple frames as references for computing optical flows eliminates such errors.
>
> **[nybp.10 - Orthogonal motion]** Thanks. We will remove “orthogonal”. Briefly, an optical flow vector can be decomposed into x and y axes. For example, the optical flow vector (3,5) has a magnitude of 3 on the +x axis and 5 on the +y axis, while a magnitude of 0 along the -x and -y axis.
>
> **[nybp.11 - Typo]** Thanks. We will fix it.
>
> **[nybp.12 - Statistical test]** Thanks. We performed the statistical tests (two-tailed t-test) in human performance between RGB-R and RGB-S. The p-value is 0.465, above 0.05. This implies that the accuracy of shuffling frames is not statistically different from reversing frames. We will revise the claim in l270. Moreover, we will perform statistical tests and report p-values for other result comparisons in the final version.
>
> **[nybp.13 - Error bars in Fig4]** Thanks. We will add the enlarged version of Fig 4 in the Appendix with error bars included.
>
> **[nybp.14 - Effect of Motion Invaraint Neurons]** We agree with the reviewer that the effect of Motion Invariant neurons is not well reflected in this ablation study. However, its effect is much more prominent when video frames are shuffled or reversed. For example, our model outperforms its ablated model without MIN in the following experiments: 62.3% vs 49.8% in RGB-R; 61.3% vs 38.0% in RGB-S, 38.7% vs 36.1% in J-6P-R, and 32.7% vs 25.5% in J-6P-S. We will emphasize this point in the final version.
>
> **[nybp.15 - A4 in Table 2]** Yes, the reviewer is correct. There is no flow snapshot neuron. We will fix it in A4, Tab 2.
>
> **[nybp.16 - Humans as upper bound]** We acknowledge that AI models can outperform humans in many tasks. However, in our BMP tasks, we argue that current AI models for action recognition remain inferior to humans across many BMP conditions, as shown in our experiments. We will clarify this point in the final version.
>
> **[nybp.17 - References]** Thanks. We will change them.
>
> **[nybp.18 - Societal impact]** Thanks. In the final version, we will expand our discussion on the societal impacts. This includes positive impacts, such as in sports and fitness; and negative impacts, such as privacy invasion and discrimination.

---

> > ### Comment · Reviewer_nybp · 2024-08-11
> >
> > I thank the authors for the detailed answer to my review and for conducting the additional experiments and analyses I'm mostly very happy with the answers. I want to mention:
> >
> > * **nybp.1 - missing AFD and two-stream models:** Thank you for including the model from Illic et al. The bad performanc of the model on the point light stimuli is interesting and demonstrates that they are are hard and powerful generalization test.
> > * **eynA.2 - train on joint or SP videos**: I very much like this additional experiment for estimating how well the architecture could solve the point light displays, but especially showing that the model generalizes surprisingly well from point light displays to RGB videos. Although this is clearly not the direction in which humans generalize, I think this result provides additional strong support for the hypothesis that the MP provides a strong and robust mechanism and makes the paper even more interesting.
> >
> > * **buWx.3-Results not supportive enough**: I agree with the reviewer that it's a bit disappointing that for the higher-frequency point light displays other methods slightly outperform MP (I would have expected other models to fail more dramatically on these stimuli). Outperforming other models even earlier would clearly be even more convincing. But I think the model performance for the more reduced stimuli and on many other stimulus categories still is strong evidence for the generalization capabilities of the model architecture, especially together with the new evidence from eynA.2.
> >
> >
> > I think the rebuttal has strengthened this paper which I consider a very strong contribution to NeurIPS: It takes inspriation from Neuroscience both in terms of mechanisms as well as datasets to build a computer vision model with high human alignment and impressive generalization capabilities, making it is a very interesting example of NeuroAI. Hence I'm increasing my rating to 8.

---

> > > ### Author Response · Authors · 2024-08-12
> > >
> > > We sincerely appreciate the time and effort you have put into thoughtfully reviewing our paper!
> > >
> > > We agree with the three points raised by the reviewer. Indeed, our BMP task is challenging. Many models including Illic et al, find the BMP dataset challenging to generalize. Our proposed model demonstrates robust generalization capabilities in biological motion perception. The results from training our model on Joint or SP videos, along with the outcomes using minimal visual information (J-6P and J-5P), further support this claim.
> > >
> > > Finally, your feedback and guidance have been helpful in enhancing our work. Thank you again for your valuable comments and support!

---

### Author Rebuttal · Authors · 2024-08-07

We appreciate all the reviewers' feedback. Results are provided in the tables here, and we encourage reviewers to refer to the PDF file containing additional figures. To differentiate these new figures and tables in the rebuttal from those in the main text, we have prefixed them with "R" in the rebuttal. For example, Fig R1 and Tab R1 correspond to Fig 1 in the rebuttal PDF and Tab 1 in this Author Rebuttal.  We have included a point-by-point response for each of the four reviewers.

**Table R1: Results of new baselines, MViT (the baseline method in our paper), and our motion perceiver (MP). Top-1 accuracy (%) is reported. Best is in bold.**
|  | RGB | J-6P | SP-8P-1LT |
|---|---|---|---|
| E2S-X3D [a] | 98.7 | 10.2 | 10.7 |
| VideoMAE [b] | 90.0 | 9.9 | 9.9 |
| MViT | **99.0** | 52.0 | 15.1 |
| TwoStream-CNN [c] | 97.0 | 15.7 | 10.4 |
| MP(ours) | 96.5 | **69.0** | **49.7** |

**Table R2:Number of trainable parameters in million (M) for baselines and our motion perceiver (MP).**
|  | ResNet3D | I3D | R(2+1)D | SlowFast | X3D | MViT | MP(ours) |
|---|:---:|:---:|:---:|:---:|:---:|:---:|:---:|
| Trainable Param (M) | 31.7 | 27.2 | 27.3 | 33.7 | 3.0 | 36.1 | 57.5 |


**Table R3: Performance of MViT, MP (block 12) and enhanced MP (blocks 1, 7 and 12) in terms of the amount of visual information. Top-1 action recognition accuracy is reported. Best is in bold.**

||RGB|J-26P|J-18P|J-14P|J-10P|J-6P|J-5P|
|-|-|-|-|-|-|-|-|
|MViT|**99.0**|76.9|75.5|74.6|68.1|52.0|47.9|
|MP(ours)|96.5|72.2|73.2|70.4|71.7|69.0|65.5|
|Enhanced MP|97.8|**80.0**|**80.6**|**78.1**|**81.0**|**78.5**|**76.7**|

**Table R4:  Human and our MP performances in per-category accuracy. Top-1 action recognition accuracy is presented in the format: Human Performance/MP Performance**.
|   | **pick up** | **throw** | **sit down** | **stand up** | **Kick something** | **Jump up** | **Point to something** | **Nod head/bow** | **Falling down** | **Arm circles** |
|:---:|:---:|:---:|:---:|:---:|:---:|:---:|:---:|:---:|:---:|:---:|
| **RGB** | 97.5/96.0 | 96.7/90.9 | 100.0/95.6 | 100.0/98.6 | 98.3/95.9 | 100.0/100.0 | 98.3/93.8 | 96.7/95.5 | 93.3/99.3 | 100.0/99.3 |
| **J-6P** | 93.3/56.4 | 75.0/76.2 | 76.7/64.2 | 85.0/70.1 | 95.0/73.2 | 95.0/91.3 | 81.7/44.7 | 91.7/68.2 | 75.0/53.4 | 95.0/93.2 |
| **SP-8P-1LT** | 81.7/49.1 | 68.3/57.0 | 80.0/57.7 | 85.0/63.7 | 90.0/28.5 | 86.7/84.1 | 66.7/4.1 | 55.0/37.7 | 96.7/58.0 | 93.3/60.1 |
| **LVI** | 66.9/44.3 | 55.3/50.5 | 50.6/45.1 | 60.3/51.7 | 78.1/26.0 | 83.6/74.2 | 40.6/3.7 | 36.1/23.4 | 89.2/48.0 | 81.7/44.1 |
| **AVI** | 95.6/53.3 | 78.9/73.2 | 81.1/67.9 | 86.7/75.0 | 95.6/73.0 | 97.8/90.7 | 83.3/44.3 | 86.7/72.0 | 80.0/50.2 | 97.8/89.9 |
| **TO** | 77.1/56.6 | 42.9/68.7 | 26.3/2.1 | 11.7/5.2 | 90.8/77.8 | 87.1/92.8 | 74.2/38.8 | 77.5/59.3 | 25.4/29.1 | 92.9/58.1 |
| **TR** | 75.4/57.2 | 41.3/36.9 | 79.2/80.5 | 85.4/83.6 | 83.8/63.8 | 60.0/27.6 | 80.4/53.4 | 72.5/67.4 | 85.8/70.7 | 64.2/12.2 |
| **ICV** | 96.7/56.3 | 80.0/76.2 | 83.3/64.2 | 85.0/70.1 | 96.7/72.9 | 96.7/91.1 | 87.2/44.4 | 85.6/68.3 | 81.1/53.4 | 95.6/93.5 |
|  |  |  |  |  |  |  |  |  |  |  |
|  |  |  |  |  |  |  |  |  |  |  |

**Table R5: Correlation values for all the models across all the BMP conditions. Best in bold.**
|  | ResNet3D | I3D | R(2+1)D | SlowFast | X3D | MViT | MP(ours) |
|:---:|:---:|:---:|:---:|:---:|:---:|:---:|:---:|
| Correlation | 0.37 | 0.53 | 0.39 | 0.71 | 0.71 | 0.66 | **0.89** |

**Table R6: Error consistency results of all the models across all the BMP conditions. Best in bold.**
|  | ResNet3D | I3D | R(2+1)D | SlowFast | X3D | MViT | MP(ours)|
|:---:|:---:|:---:|:---:|:---:|:---:|:---:|:---:|
| error consistency | 0.135 | 0.175 | 0.110 | 0.236 | 0.209 | 0.219 | **0.240** |

**Reference list:**

[a] Filip et al., Is appearance free action recognition possible? ECCV 2022.

[b] Zhan et al., Videomae: Masked autoencoders are data-efficient learners for self-supervised video pre-training. NeurIPS 2022.

[c] Karen et al., Two-stream convolutional networks for action recognition in videos. NeurIPS 2014.

[d] Yun et al., Cutmix: Regularization strategy to train strong classifiers with localizable features. ICCV 2019.

[e] Dabouei et al., Supermix: Supervising the mixing data augmentation. CVPR 2021.

[f] Hong  et al., Stylemix: Separating content and style for enhanced data augmentation. CVPR 2021.

[g] Zhong et al., Random erasing data augmentation. AAAI 2020.

[h] Kim et al., Learning temporally invariant and localizable features via data augmentation for video recognition. ECCV 2020 Workshops.

[i] Kowal et al., A deeper dive into what deep spatiotemporal networks encode: Quantifying static vs. dynamic information. CVPR 2022.

[j] Choi et al., Why can't i dance in the mall? learning to mitigate scene bias in action recognition. NeurIPS 2019.

[k] He et al., Human action recognition without human. ECCV 2016 Workshops.

[l] Yamins et al., Performance-optimized hierarchical models predict neural responses in higher visual cortex. PNAS 2014.

[m] Zhang et al., Finding any Waldo with zero-shot invariant and efficient visual search. Nat. Commun.2018.

[n] Geirhos et al., Generalisation in humans and deep neural networks. NeurIPS 2018.

[o] Schrimpf et al. Integrative benchmarking to advance neurally mechanistic models of human intelligence. Neuron 2020.

[p] Geirhos et al., Beyond accuracy: quantifying trial-by-trial behaviour of CNNs and humans by measuring error consistency. NeurIPS 2020.

[q] Simonyan et al., Two-stream convolutional networks for action recognition in videos. NeurIPS 2014.

---

### Decision · Program_Chairs · 2024-09-25

**Decision:**

Accept (poster)

**Comment:**

All reviewers agreed that this paper is suitable for publication. The authors used a classic psychophysics paradigm in human vision to motivate the development of a new vision architecture. The problem and approach will be inspiring to cognitive scientists, and the benchmark will be useful for cognitive scientists and potentially the computer vision community to better gauge the alignment (or lack thereof) of there models with human vision.